# Molecular Dynamics Simulations of Liposomes: Structure, Dynamics, and Applications

**DOI:** 10.3390/membranes15090259

**Published:** 2025-08-29

**Authors:** Ehsan Khodadadi, Ehsaneh Khodadadi, Parth Chaturvedi, Mahmoud Moradi

**Affiliations:** Department of Chemistry and Biochemistry, University of Arkansas, Fayetteville, AR 72701, USA; ehsank@uark.edu (E.K.); ekhodada@uark.edu (E.K.); parthc@uark.edu (P.C.)

**Keywords:** liposomes, coarse-grained molecular dynamics, MARTINI force field, membrane biophysics, lipid vesicle curvature, PEGylation, cholesterol partitioning, drug delivery nanocarriers, bilayer remodeling, interleaflet coupling

## Abstract

Liposomes are nanoscale, spherical vesicles composed of phospholipid bilayers, typically ranging from 50 to 200 nm in diameter. Their unique ability to encapsulate both hydrophilic and hydrophobic molecules makes them powerful nanocarriers for drug delivery, diagnostics, and vaccine formulations. Several FDA-approved formulations such as Doxil^®^ (Baxter Healthcare Corporation, Deerfield, IL, USA), AmBisome^®^ (Gilead Sciences, Inc., Foster City, CA, USA), and Onivyde^®^ (Ipsen Biopharmaceuticals, Inc., Basking Ridge, NJ, USA) highlight their clinical significance. This review provides a comprehensive synthesis of how molecular dynamics (MD) simulations, particularly coarse-grained (CG) and atomistic approaches, advance our understanding of liposomal membranes. We explore key membrane biophysical properties, including area per lipid (APL), bilayer thickness, segmental order parameter ($S_{CD}$), radial distribution functions (RDFs), bending modulus, and flip-flop dynamics, and examine how these are modulated by cholesterol concentration, PEGylation, and curvature. Special attention is given to curvature-induced effects in spherical vesicles, such as lipid asymmetry, interleaflet coupling, and stress gradients across the leaflets. We discuss recent developments in vesicle modeling using tools such as TS2CG, CHARMM-GUI Martini Maker, and Packmol, which have enabled the simulation of large-scale, compositionally heterogeneous systems. The review also highlights simulation-guided strategies for designing stealth liposomes, tuning membrane permeability, and enhancing structural stability under physiological conditions. A range of CG force fields, MARTINI, SPICA, SIRAH, ELBA, SDK, as well as emerging machine learning (ML)-based models, are critically assessed for their strengths and limitations. Despite the efficiency of CG models, challenges remain in capturing long-timescale events and atomistic-level interactions, driving the development of hybrid multiscale frameworks and AI-integrated techniques. By bridging experimental findings with in silico insights, MD simulations continue to play a pivotal role in the rational design of next-generation liposomal therapeutics.

## 1. Introduction

Liposomes are nanoscale vesicles composed of one or more concentric lipid bilayers enclosing an aqueous core [1,2]. Their amphiphilic architecture allows for the simultaneous encapsulation of hydrophilic and hydrophobic therapeutic agents, including small molecules, peptides, proteins, nucleic acids, and imaging compounds. This versatility, coupled with favorable biocompatibility and tunable physicochemical properties, has made liposomes one of the most extensively used nanocarriers in modern medicine [3,4].

Clinically, liposomes have enabled a new generation of targeted drug delivery systems. Several formulations have gained FDA approval, such as Doxil^®^ (liposomal doxorubicin), AmBisome^®^ (liposomal amphotericin B), and Onivyde^®^ (liposomal irinotecan), offering improved pharmacokinetics, reduced systemic toxicity, and enhanced therapeutic efficacy [5,6,7]. These successes underscore the importance of understanding how bilayer composition and architecture influence liposome function in vivo.

Despite these advances, the clinical translation of liposomal formulations has also encountered significant challenges. For instance, ThermoDox^®^ (Celsion Corporation, Lawrenceville, NJ, USA) , a lyso-thermosensitive liposomal formulation of doxorubicin designed to release its payload upon localized heating, failed to meet primary endpoints in large-scale clinical trials, including the Phase III OPTIMA trial for hepatocellular carcinoma [8]. Contributing factors included inadequate thermal control, heterogeneous tumor perfusion, and insufficient drug release at target sites. Similarly, Mikhail et al. [9] demonstrated that although ThermoDox showed promise in preclinical bladder cancer models, clinical efficacy remains limited. These outcomes highlight the need for a deeper mechanistic understanding of liposome–tissue interactions and improved simulation models that incorporate physiological complexity.

Biological behavior and therapeutic performance of liposomes are governed by the biophysical properties of their lipid bilayers. Membrane rigidity, thickness, fluidity, surface charge, and curvature directly affect drug loading, stability, release kinetics, and interactions with cells and biological fluids [10,11,12]. For example, cholesterol is a key regulator of bilayer order and thickness, modulating membrane stiffness and drug partitioning [13,14]. One of the key insights provided by CG-MD is the effect of curvature on membrane dynamics, such as lipid flip-flop, interleaflet coupling, and lateral lipid redistribution. Meanwhile, membrane curvature in small unilamellar vesicles generates lateral pressure gradients between leaflets, influencing lipid redistribution, interleaflet coupling, and lipid flip-flop phenomena essential to cellular uptake, endocytosis, and membrane fusion [15,16].

To improve circulation time and reduce immune recognition, polyethylene glycol (PEG) chains are often grafted onto the liposome surface. This PEGylation provides a steric barrier that enhances systemic stability and promotes tumor accumulation via the enhanced permeability and retention (EPR) effect [17]. However, PEG chains also influence membrane dynamics and local curvature, potentially affecting drug encapsulation and release profiles [18].

Despite advances in experimental techniques such as cryo-electron microscopy, small-angle X-ray scattering (SAXS), and fluorescence microscopy, capturing fast, transient, or rare molecular events in liposomes, such as lipid flip-flop, interleaflet stress relaxation, or curvature-induced lipid redistribution, remains challenging [19,20]. MD simulations, particularly CG approaches, provide powerful tools to bridge this resolution gap by enabling microsecond-scale exploration of membrane structure and dynamics [21,22].

The MARTINI force field [23,24,25] is the most widely used CG model for simulating lipid membranes. Its 4:1 mapping strategy reduces system complexity while preserving essential interactions. Figure 1 illustrates the CG mapping strategy used for common lipid molecules like DPPC, cholesterol, and benzene, highlighting how chemical structures are translated into MARTINI bead types with appropriate hydrophilicity or ring specificity. MARTINI 2 accurately reproduces bilayer thickness, area per lipid, lateral diffusion, and phase behavior, while MARTINI 3 improves chemical specificity and electrostatics [26,27]. Other CG frameworks, such as SPICA [28,29], SIRAH [30,31], ELBA [32,33], and SDK extend applicability to proteins, charged systems, and polarizable interactions. Recently, machine learning-based CG approaches have emerged as flexible, data-driven alternatives for developing accurate interaction potentials [34,35,36].

CG-MD simulations have offered crucial insights into the effects of cholesterol on lipid packing, ordering, and phase behavior, as well as how curvature alters interleaflet stress and lipid asymmetry. These structural factors govern key biological processes such as membrane fusion, pore formation, and protein insertion, all of which are relevant to liposomal function [11,15,37]. Furthermore, MD simulations help interpret drug lipid interactions and optimize formulation strategies. For instance, Siani et al. [38] simulated doxorubicin-loaded membranes and showed how lipid composition and cholesterol levels modulate drug penetration and retention (Figure 2).

In this review, we critically examine the utility of MD simulations in elucidating structural and dynamic properties of liposomes, with emphasis on cholesterol-mediated effects, membrane curvature, PEGylation, and drug interactions. We evaluate the capabilities and limitations of different CG and atomistic force fields, including MARTINI, SPICA, SIRAH, ELBA, and ML-based models, in capturing biologically relevant membrane phenomena. Our goal is to guide rational liposome design for improved performance in therapeutic and diagnostic applications [39].

## 2. Biophysical Role of Cholesterol in Liposomes

Cholesterol is a key regulatory molecule in liposomal membranes. Its amphipathic nature and rigid tetracyclic ring structure allow it to insert between phospholipid tails, thereby modulating membrane fluidity, thickness, permeability, and mechanical stability properties critical to liposomal drug delivery systems [40,41,42]. CG and atomistic MD simulations, using force fields like MARTINI, SPICA, SIRAH, and ELBA, have deepened our understanding of cholesterol’s multifaceted impact on bilayer behavior.

### 2.1. Structural Effects and Bilayer Ordering

Cholesterol enhances acyl chain ordering by occupying interstitial spaces between lipid tails. This condensing effect reduces the area per lipid (APL), increases bilayer thickness, and decreases lateral diffusion [6,11].

Experimental studies have consistently demonstrated that cholesterol increases bilayer thickness by ordering lipid tails and reducing free volume. For instance, Tristram-Nagle et al. [43] showed using X-ray diffraction that the thickness of DPPC bilayers increases from approximately 36 Å to 44 Å with rising cholesterol content. Similarly, Hung et al. [44] employed neutron scattering to confirm this thickening in DOPC–cholesterol systems. These findings align well with atomistic MD simulations, such as those based on the CHARMM force field and coarse-grained MARTINI simulations, both of which report bilayer thickening in the range of 5–10 Å, depending on cholesterol concentration. This agreement underscores the predictive capability of MD approaches in capturing cholesterol-mediated biophysical changes in lipid membranes.

### 2.2. Phase Behavior and Domain Formation

Cholesterol responds to membrane curvature, especially in vesicles or tubules. It promotes the formation of non-lamellar structures like inverted hexagonal (H_II_) phases, which are relevant in endosomal escape and fusion events [45]. These behaviors emphasize cholesterol’s capacity to regulate liposome flexibility and morphological adaptability during biological transport.

### 2.3. Drug Encapsulation and Controlled Release

Structurally, cholesterol lowers bilayer permeability by reducing membrane defects and stabilizing the hydrophobic core. This contributes to improved encapsulation efficiency and sustained drug release, especially under external stimuli like heat or ultrasound [10,46]. Cholesterol also decreases the phase transition temperature ($T_{m}$), enhancing bilayer fluidity under physiological conditions while maintaining structural integrity [47,48]. When combined with stabilizing agents like elastin-like recombinamers, cholesterol enhances the shelf life and responsive behavior of liposomal formulations [49]. While these encapsulation properties depend heavily on bilayer composition, the physical shape and curvature of the membrane further modulate liposomal performance. These geometric features are explored next.

### 2.4. Membrane Curvature Effects on Lipid Behavior

Cholesterol alters lipid headgroup orientation and local pressure profiles, affecting solute partitioning across the bilayer. COSMOmic and MD studies reveal orientation-dependent effects that impact drug localization, further highlighting cholesterol’s functional significance in liposome design [50,51].

Beyond composition, the curvature and geometry of lipid vesicles play a pivotal role in modulating their structural and dynamic properties. Liposomes exhibit a wide range of curvatures depending on their size and shape, and these geometric differences have profound consequences on membrane packing, lipid tail ordering, lateral diffusion, flip-flop dynamics, and domain formation. Understanding these curvature-induced effects is essential for the rational design of liposomal drug delivery systems, especially when transitioning from planar model membranes to realistic three-dimensional vesicle geometries.

MD simulations have provided valuable insights into how curvature modulates the segmental order parameter (S_CD_), a key indicator of lipid tail ordering. Studies by Nishizawa et al. [52] and Domanska et al. [16] demonstrated that increased membrane curvature leads to reduced lipid tail order, particularly in the inner leaflet of small vesicles. This effect is attributed to geometrical stress and tighter molecular packing on the concave side of the bilayer, as also evidenced by changes in local area density (Figure 3). Such changes reduce chain alignment and enhance flexibility, critical parameters for drug retention and controlled release.

These structural variations can be visualized through simulations of lipid area density under curvature stress. As shown in Figure 3, the local packing density of POPC bilayers changes significantly with increasing curvature, emphasizing how membrane geometry impacts bilayer structure and drug encapsulation potential.

Curvature also influences membrane tension and bending rigidity. Gu et al. [15] showed that curvature-dependent deformation and shape fluctuations are directly linked to bending modulus and Gaussian curvature, which are modulated by lipid composition and cholesterol content. Smaller vesicles with higher curvature exhibit increased bending energy and altered diffusion profiles, which can impact their fusion behavior and stability in vivo.

Importantly, curvature can drive the formation of lipid microdomains and regulate transbilayer lipid migration. Asymmetrical curvature across bilayer leaflets introduces mechanical stress that can modulate lipid phase behavior and promote flip-flop events. This phenomenon was investigated by Hakobyan and Heuer [20], who identified key molecular factors responsible for raft-like domain formation under curved conditions. Parthasarathy and Groves [53] further emphasized how curvature-induced heterogeneity contributes to lateral sorting of membrane components, affecting surface presentation of ligands and encapsulated drug distribution.

These curvature-related effects also extend to lipid–protein interactions. Brown [54] discussed how membrane curvature can generate forces that influence protein conformations and localization factors that must be considered in the context of functionalized liposomes carrying targeting peptides or antibodies. Additionally, Larsen [55] provided a comprehensive MD-based framework for simulating curved membranes, confirming that vesicle size and geometry regulate mechanical and diffusive properties at multiple scales.

One important dynamic process regulated by curvature is transbilayer lipid movement, or flip-flop, which we discuss below.

#### Lateral Diffusion and Flip-Flop Dynamics

Membrane curvature exerts a profound influence on lipid bilayer dynamics, particularly in facilitating transbilayer movement of lipids (flip-flop) and enhancing interleaflet coupling. Lipid flip-flop, the translocation of molecules like phospholipids or cholesterol between the two leaflets of a bilayer, is typically an energetically unfavorable event due to the need to traverse the hydrophobic membrane core. However, curvature-induced stress and asymmetry can reduce this energetic barrier, making flip-flop more prevalent, particularly in coarse-grained simulations that capture long-timescale processes.

Róg et al. [11] demonstrated that cholesterol’s unique structure enables it to orient itself in ways that reduce hydrophobic mismatch, promoting dynamic movement across leaflets. They emphasized the condensing and ordering effects of cholesterol in both planar and curved geometries, effects that are enhanced by sterol tilt and asymmetrical packing under curvature stress. Complementing this, de Meyer and Smit [13] used a mesoscopic model to reveal how cholesterol induces a non-ideal mixing behavior in phospholipid membranes, termed the “condensation effect”. Their results suggest that cholesterol decreases the area per lipid while enhancing bilayer thickness properties that help alleviate curvature-induced mechanical strain.

Gu et al. [15] provided atomistic insight into cholesterol flip-flop behavior within heterogeneous membranes. Their simulations showed that cholesterol flip-flop rates are highly dependent on the local lipid environment, with Lo and Ld domains differentially affecting cholesterol orientation and flip-flop frequency. Notably, curvature and lipid heterogeneity were shown to synergistically facilitate transleaflet migration, which may play a critical role in maintaining membrane homeostasis under dynamic conditions.

Feller and MacKerell [37] added further depth by demonstrating how cholesterol acts as a mechanical buffer, adjusting its positioning in response to local stress and promoting bilayer stability through its interactions with both saturated and unsaturated lipid tails. Their empirical potential energy function accurately modeled the dynamic tension within bilayers and highlighted cholesterol’s role in modulating leaflet asymmetry under stress.

Additionally, studies by Grouleff et al. [56] and Alavizargar et al. [57] offered mechanistic explanations of how cholesterol stabilizes interleaflet coupling through its ordering influence on surrounding lipids. Grouleff et al. emphasized the ability of cholesterol to synchronize structural changes across the bilayer, thereby affecting protein conformation and function. Alavizargar et al. showed that cholesterol’s high planarity and its asymmetric interactions with lipid tails enhance packing efficiency and may indirectly promote lipid redistribution, especially under curvature-induced tension.

Taken together, these studies converge on a compelling picture: membrane curvature enhances lipid flip-flop by creating asymmetries in lateral pressure and molecular orientation, while cholesterol dynamically responds to and modulates these changes. The coupling of flip-flop dynamics with interleaflet interactions contributes to the stabilization of curved structures like liposomes and supports functional membrane remodeling. Incorporating these curvature-dependent behaviors into MD-based liposomal models is essential for accurately capturing lipid redistribution, bilayer asymmetry, and stress responses. These factors are critical for optimizing liposome-based drug delivery systems.

### 2.5. Cholesterol Regulation via Cyclodextrins

Cholesterol induces liquid-ordered (Lo) phase formation at moderate concentrations, stabilizing the coexistence of Lo and liquid-disordered (Ld) domains [19]. This nanoscale heterogeneity enables spatial segregation of lipid species and supports membrane protein localization. Simulations reveal that cholesterol drives phase separation by modulating lipid-lipid interaction energies and promoting raft-like structures, particularly in systems containing saturated lipids such as DPPC or DSPC [6,18].

### 2.6. Solute Partitioning and Headgroup Orientation

Cyclodextrins (CDs), particularly methyl-β-cyclodextrin (MβCD), modulate cholesterol levels in experimental systems. Atomistic studies show that CDs can spontaneously extract cholesterol by forming 2:1 CD:cholesterol complexes aligned perpendicular to the bilayer [58]. This extraction perturbs local lipid packing and influences membrane permeability, making CDs useful tools for both research and therapy.

### 2.7. Influence of Unsaturation and Headgroup Chemistry

Simulations show that cholesterol restricts lipid tail motion, reducing lateral diffusion while increasing ordering [11]. Its effects on flip-flop rates are more nuanced—cholesterol may hinder phospholipid translocation but increase its own flip-flop due to its small headgroup and structural rigidity. These dynamics are crucial for understanding membrane asymmetry and lipid redistribution.

The architecture, dynamics, and function of liposomes are intricately governed by the chemical and structural diversity of their lipid components. Variations in phospholipid tail saturation, headgroup identity, and cholesterol content critically determine the biophysical behavior of lipid bilayers. These compositional parameters influence bilayer fluidity, lateral diffusion, bending rigidity, membrane thickness, and spontaneous curvature. CG-MD simulations offer a robust platform to systematically study these effects at both the molecular and mesoscopic scales. They provide insights into how specific lipid compositions modulate liposome morphology, curvature stress, and vesicle stability across a range of biologically relevant conditions [23,24,25,59].

The level of unsaturation in lipid acyl chains is another determinant of membrane architecture. Unsaturated lipids like dioleoylphosphatidylcholine (DOPC) and dioleoylphosphatidylethanolamine (DOPE) possess one or more cis-double bonds in their tails, creating kinks that disrupt tight packing. This leads to increased membrane fluidity, decreased bilayer thickness, and enhanced lateral diffusion characteristics favorable for membrane remodeling and dynamic vesicle behavior [17].

Simulations by Klauda et al. and others have shown that DOPC bilayers exhibit high tail disorder and are prone to curvature, while fully saturated lipids, such as dipalmitoylphosphatidylcholine (DPPC), form gel-phase bilayers that are highly ordered, with a greater bending modulus and lower permeability. Mixing saturated and unsaturated lipids allows for fine-tuning bilayer mechanical properties and curvature stress, enabling adaptive design of flexible or robust liposomes depending on application [2,19].

In addition to saturation, the lipid headgroup chemistry plays a pivotal role in determining bilayer packing and curvature. Phosphatidylethanolamine (PE), with its small zwitterionic headgroup, induces negative curvature and promotes non-lamellar structures, making it suitable for fusion or delivery systems requiring rapid vesicle deformation. In contrast, phosphatidylcholine (PC), with a bulkier headgroup, stabilizes lamellar structures with minimal spontaneous curvature [48,60].

Recent CG-MD simulations have shown that increasing the PE content in a bilayer can drive tubulation, budding, and vesicle formation due to the associated curvature stress. These findings highlight the utility of headgroup engineering in vesicle design, especially when targeting membrane fusion or endosomal escape [61,62].

## 3. Foundations of Liposome Simulation and Biophysical Insights

This section is divided into two complementary parts. First, we present the methodological foundations of CG-MD, with a focus on force fields, mapping schemes, and simulation frameworks. Second, we explore the application of CG-MD to liposome systems by evaluating biophysical descriptors that enable mechanistic insights into liposome structure and function.

### 3.1. Principles of CG Liposome Modeling

MD simulations have become indispensable tools in membrane biophysics, offering atomistic or mesoscopic insight into the structural and dynamical properties of biomolecular systems. By solving Newton’s equations of motion for a system of interacting particles, MD simulations allow researchers to capture the conformational behavior and thermodynamic fluctuations of complex assemblies, such as lipid bilayers, liposomes, and membrane-bound proteins [21,22,23].

In liposome and bilayer research, MD simulations provide mechanistic detail on a wide range of phenomena, including lipid packing, bilayer thickness, lateral diffusion, curvature generation, lipid flip-flop, and domain formation [11,15]. Moreover, MD techniques support the interpretation and extension of experimental data derived from SAXS, NMR, and cryo-EM, bridging the gap between nanoscopic dynamics and observable macroscopic properties.

Two primary classes of MD simulations are typically employed in membrane modeling: all-atom (AA) and CG. AA simulations, while highly detailed and capable of resolving hydrogen bonding networks, hydration shells, and electrostatic interactions, are limited in both spatial and temporal resolution due to computational cost. Typically, systems are constrained to several hundred nanometers and simulation timescales up to microseconds [27]. A detailed comparison between AA and CG simulations is presented in Table 1.

To access biologically relevant length and timescales, CG simulations simplify molecular systems by grouping atoms into larger interaction beads. This abstraction allows for reduced degrees of freedom while retaining key physicochemical features. The most widely adopted CG model is the MARTINI force field, originally developed by Marrink et al. [23]. MARTINI 2 introduced a four-to-one mapping scheme that enables simulation of lipid membranes, membrane remodeling, protein–lipid interactions, and vesicle fusion across microsecond to millisecond timescales with excellent computational efficiency.

A major validation of MARTINI’s accuracy was demonstrated by Monticelli et al. [24], who modeled the insertion of KALP peptides into DLPC bilayers. Their work confirmed that CG simulations could replicate tilt angles and hydrophobic mismatch effects seen in AA simulations and experiments (Figure 4), establishing MARTINI as a robust platform for membrane protein modeling.

Building upon this foundation, MARTINI 3 was introduced by Souza et al. [25] to enhance the force field’s fidelity in representing hydrogen bonding, stacking interactions, and polarity. Notably, MARTINI 3 improves the accuracy of free energy profiles, radial distribution functions, and molecular orientation parameters. Its expanded chemical resolution also enables more realistic modeling of nucleic acids, saccharides, and charged lipids in multicomponent systems, supporting applications in membrane biology and drug delivery (Figure 5).

In addition to MARTINI, alternative CG force fields such as SPICA [28,29], SIRAH [30,31], and ELBA [32] offer complementary strengths. SPICA models emphasize membrane thickness and protein–lipid interfaces; SIRAH integrates protein and nucleic acid dynamics in explicit water environments; and ELBA uses a hybrid electrostatics approach to enhance polar interaction resolution. These force fields expand the toolbox for membrane researchers aiming to investigate diverse systems such as ion channels, PEGylated vesicles, and protein–lipid assemblies [5,62,63].

Having outlined the major CG force fields, we now turn to the structural and mechanical parameters typically extracted from CG simulations to characterize liposome behavior.

### 3.2. Biophysical Observables from CG-MD

Modern MD workflows frequently compute structural and mechanical properties from trajectories to quantify membrane behavior under different lipid compositions, curvatures, and external stimuli. Key parameters include the following:**APL:** APL is defined as the average surface area each lipid occupies in a leaflet and is a critical descriptor of lipid packing density and membrane phase state [41,57]. A decrease in APL is associated with tighter lipid packing (e.g., due to cholesterol condensation), whereas an increase may signal disorder, increased curvature, or bilayer stress. Changes in APL are often linked to membrane permeability and interactions with proteins or polymers. Curved membranes or asymmetric lipid distribution can cause differences in APL between leaflets, revealing the impact of local geometry on lipid organization.**Bilayer thickness:** Bilayer thickness is typically quantified as the average distance between the phosphate headgroups (P–P) of opposing leaflets, offering a convenient and widely used metric for membrane thickness [10,16]. However, alternative definitions based on the distance between specific carbon atoms in the glycerol backbone, such as C1–C1 or C2–C2 distances, are also employed, particularly in atomistic simulations [64,65]. These definitions can yield slightly different values and are chosen based on the resolution of the model and the structural detail of interest. Bilayer thickness reflects the degree of acyl chain extension and lipid ordering. Cholesterol typically increases bilayer thickness by promoting tail alignment and ordering, while unsaturated lipids or high membrane curvature can reduce it by introducing kinks and lateral pressure asymmetries. Thickness also varies with temperature, PEGylation, and lipid composition, and plays a crucial role in determining hydrophobic mismatch with transmembrane proteins or synthetic nanoparticles. Furthermore, spatial variations in bilayer thickness can be indicative of phase separation, lipid domain formation, or the presence of fusion intermediates in dynamic membrane processes.**Order parameter (S_CD_):** The order parameter, commonly denoted as $S_{CD}$, is a widely used metric derived from deuterium NMR experiments and molecular dynamics simulations to quantify the orientational ordering of lipid acyl chains [11,14]. Higher $S_{CD}$ values correspond to more ordered, extended lipid tails, typically observed in cholesterol-rich or gel-phase membranes, while lower values indicate increased tail flexibility and fluidity. This parameter is particularly useful for assessing how components such as embedded drugs or PEGylated lipids influence local membrane dynamics. Spatial profiles of $S_{CD}$ across the bilayer reveal differences between Sn1 and Sn2 chains, chain tilt, lipid tail disorder, and curvature-induced variations in membrane structure.**RDF:** RDF describes how density varies as a function of distance from a reference molecule [37,66]. In lipid systems, RDFs help quantify lipid–lipid, lipid–cholesterol, or lipid–protein interactions, identifying clustering, local organization, and hydration patterns. They provide a statistical view of local molecular arrangement and can reveal phase behavior transitions and lipid domain formation in multicomponent systems.**Bending modulus and curvature stress:** The bending modulus ($K_{c}$) quantifies membrane stiffness and the energetic cost of bending [2,21]. It governs how easily membranes can deform during vesicle formation, fusion, or budding. Curvature stress arises from leaflet asymmetry, cholesterol partitioning, or protein insertion and determines the membrane’s ability to accommodate shape changes under stress. Recent modeling work highlights that in asymmetric membranes, particularly those with cholesterol flip-flop differential stress, defined as an imbalance in lateral stress between leaflets, critically influences curvature and bending energy [67,68]. This concept adds a mechanistic layer to our understanding of curvature stress, especially relevant for dynamically heterogeneous vesicle systems.

By enabling long-timescale modeling of lipid assemblies, CG simulations are uniquely suited to explore liposome formation, PEGylation effects, fusion intermediates, and encapsulated drug behavior in complex membrane geometries. Tools such as Triangulated surface mesh to generate CG lipid structures (TS2CG) [69], CHARMM-GUI Martini Maker [70], and Packmol [71] facilitate system construction and support multiscale modeling of large vesicles. These advances are essential for rational design of lipid-based drug delivery systems where structure–function relationships depend critically on membrane mechanics, curvature, and composition.

These biophysical metrics provide a foundation for understanding the dynamic behavior of liposomes across varying compositions, geometries, and external stimuli. In particular, they offer mechanistic insights into how structural modifications, such as cholesterol enrichment, PEGylation, or the inclusion of unsaturated lipids, modulate membrane fluidity, curvature elasticity, and domain formation, ultimately influencing the efficacy of liposome-based drug delivery systems.

The curvature inherent to spherical vesicles introduces unique mechanical challenges compared to planar bilayers. In small unilamellar vesicles (SUVs) or highly curved lipid systems, curvature can induce leaflet asymmetry, interdigitation, and altered packing density [16,52]. These effects influence not only APL and bilayer thickness but also promote spontaneous lipid flip-flop and enhance the mixing of inner and outer leaflet components. Such curvature-driven dynamics are particularly relevant in the context of PEGylated and stimuli-responsive liposomes, which often operate under dynamic mechanical conditions in physiological environments [10].

Recent advancements in CG-MD, especially with MARTINI 3, enable the modeling of large vesicle systems with realistic levels of curvature and asymmetric lipid distribution. The combination of CG-MD with tools like TS2CG [69] and CHARMM-GUI’s Martini Maker [70] has facilitated the generation of vesicles up to hundreds of nanometers in diameter. These systems capture slow dynamic processes such as lipid rearrangement, pore formation, and curvature adaptation that are inaccessible to atomistic simulations.

Incorporating PEG chains into liposomes has become a central strategy for extending circulation time and reducing immune recognition. Simulations have shown that PEGylation not only modifies surface hydration and steric repulsion but also impacts local curvature and membrane order [72]. PEG–lipid incorporation often results in a local thinning of the bilayer and reduction in $S_{CD}$, which can alter fusion potential and cargo release kinetics. These structural modifications are critical for designing long-circulating stealth liposomes and stimuli-responsive nanocarriers [62].

Moreover, simulation studies have revealed that cholesterol acts as a molecular regulator of membrane properties, with concentration-dependent effects. At moderate concentrations (20–30%), cholesterol orders lipid tails and reduces APL, enhancing membrane rigidity and lowering permeability [13,15]. When cholesterol reaches or exceeds 33% molar fraction, particularly in ternary systems like DOPC:Chol:Sphingomyelin (1:1:1), it can promote lateral phase separation into coexisting liquid-ordered (Lo) and liquid-disordered (Ld) domains [57]. This phase coexistence underlies lipid raft formation, affecting protein partitioning and signaling platform organization. Additionally, interleaflet coupling and flip-flop events, especially for cholesterol and DOPC, are strongly modulated by curvature, leaflet stress, and asymmetric lipid composition. As demonstrated in recent CG-MD studies [11,13,15], vesicle curvature lowers the energy barrier for lipid translocation, particularly for small amphiphilic molecules like cholesterol. This dynamic redistribution affects not only membrane composition and order but also the mechanical properties of the vesicle, which are crucial for understanding membrane remodeling during fusion, endocytosis, or nanoparticle–cell interactions.

The interplay of these structural descriptors, force fields, and modeling tools underscores the value of CG-MD simulations in rationalizing the behavior of liposomes under physiological and synthetic conditions. By systematically tuning membrane composition, curvature, and functionalization, researchers can design liposomal systems with tailored mechanical stability, permeability, and targeting efficiency for diverse biomedical applications. The following case studies illustrate how these descriptors and curvature concepts are applied in realistic, fully enclosed vesicle models.

### 3.3. Modeling Entire Liposomes: CG and Atomistic Perspectives

While many simulation studies focus on planar bilayers or membrane patches, modeling entire liposomes has become increasingly important for capturing curvature-driven effects and realistic mechanical behavior. Full vesicle simulations allow for direct investigation of lipid packing asymmetry, local membrane tension, and spatially varying bilayer properties, phenomena that are difficult to resolve in flat systems. Braun et al. [64] employed the MARTINI CG force field to simulate a complete 34 nm DOPC vesicle, providing detailed insights into bilayer thickness variations, curvature-induced stress, and spatial heterogeneity in mechanical properties. Their analysis revealed how local curvature modulates bilayer elasticity and structural organization across the vesicle surface. In a complementary approach, Drabik et al. [65] performed all-atom molecular dynamics simulations using the CHARMM force field to investigate the structural and mechanical properties of smaller (20 nm) vesicles composed of DPPC, DSPC, and other phospholipids. By simulating pressure-induced deformation, they extracted curvature-dependent bilayer responses and demonstrated how lipid composition influences vesicle stiffness and thickness. These full-vesicle simulations, spanning both coarse-grained and atomistic regimes, represent a critical step toward connecting molecular-level lipid behavior with the mesoscale architecture and mechanical function of clinically relevant liposomes. Incorporating such approaches enhances our ability to design stable, deformable, and tunable drug delivery vehicles that function under physiologically relevant conditions.

### 3.4. Simulation Frameworks, Force Fields, and Validation

The choice of force field in MD simulations is central to accurately capturing the structure, dynamics, and function of liposomes. In CG simulations, where multiple atoms are grouped into single interaction beads, several force fields have been developed and refined to balance computational efficiency with biophysical accuracy. This section outlines and compares the most commonly used CG force fields in liposome modeling, including their physical basis, applications, and limitations.

#### 3.4.1. The MARTINI Force Field Family

The MARTINI force field, originally developed by Marrink et al. [23], has become the cornerstone of CG-MD simulations for biomolecular systems. By mapping multiple atoms into single interaction beads, MARTINI enables simulations of large-scale systems while maintaining an adequate level of structural and dynamic accuracy. Its evolution over time has yielded multiple versions and variants, each aimed at improving the representation of biomolecular interactions.

#### 3.4.2. Wet MARTINI: MARTINI 2 and MARTINI 3

Wet MARTINI incorporates explicit solvent beads via a 4:1 mapping scheme, where four water molecules are represented by one bead. This configuration makes it suitable for modeling hydration-dependent phenomena such as membrane fusion, lipid–protein binding, solute permeation, and domain formation.

##### MARTINI 2

MARTINI 2, introduced by Marrink et al. [23] and extended to proteins by Monticelli et al. [24], marked a breakthrough in CG lipid modeling. It was systematically validated against experimental data for APL, bilayer thickness, lateral diffusion coefficients, and lipid phase behavior. For example, in DOPC bilayers, MARTINI 2 predicts APL values of ∼0.64–0.66 nm^2^ and bilayer thicknesses of ∼3.5–3.6 nm, in good agreement with neutron scattering and X-ray diffraction data. Although the model accurately reproduces lipid diffusion trends, it tends to overestimate absolute diffusion coefficients due to its relatively soft potential energy surfaces. Limitations of MARTINI 2 include a lack of resolution for directional interactions such as hydrogen bonds and cation–π interactions, making it less suited for systems with polar or structured water environments.

##### MARTINI 3

MARTINI 3, released by Souza et al. [25] and extended to small molecules by Alessandri et al. [26], introduces a reparameterized framework with more than 100 distinct bead types. This allows for improved representation of aromatic, heterocyclic, charged, and polar chemical functionalities. The model enhances the reliability of simulations involving DNA, RNA, intrinsically disordered proteins, glycolipids, and nanoparticles. MARTINI 3 features anisotropic bead types and refined nonbonded interaction potentials to mitigate issues such as overbinding observed in earlier versions. It supports improved cross-compatibility with polarizable water models and crowding agents. Validation benchmarks show enhanced agreement with experimental free energies of solvation, partitioning behavior, and electrostatic screening. Despite these advancements, MARTINI 3 still exhibits the typical CG drawback of accelerated dynamics, necessitating time rescaling. It also lacks explicit treatment of hydrogen bond geometries and dipole alignment, which limits its accuracy in modeling directionally sensitive interactions.

#### 3.4.3. Dry MARTINI

Dry MARTINI eliminates explicit solvent beads entirely, using rescaled non-bonded interaction parameters to maintain membrane structure. This version significantly increases simulation speed up to an order of magnitude, allowing for the study of large-scale systems, such as vesicles exceeding 100 nm in diameter and viral envelopes, over extended timescales. Dry MARTINI is especially effective for studying vesicle self-assembly, large-scale membrane deformation, and drug encapsulation processes. However, the absence of solvent removes hydration-mediated effects, including water-induced repulsion, ion screening, and solute permeation mechanisms. Consequently, it is unsuitable for investigating fusion events, pore formation, or other phenomena where water plays a functional role. Although Dry MARTINI can be tuned to reproduce structural metrics like bending rigidity and APL, it underestimates undulation amplitudes and hydration forces critical for accurate modeling of protein–lipid interfaces [73].

#### 3.4.4. MARTINI 2.3P: A Polarizable Variant

To improve the representation of electrostatic interactions and specific contacts such as choline–aromatic cation–π interactions, Khan et al. [74] introduced MARTINI 2.3P. This variant employs specially parameterized polarizable beads and modifies the Lennard-Jones interaction matrix to better reproduce the binding of peripheral proteins to PC-rich membranes. Simulations using MARTINI 2.3P showed improved accuracy in membrane orientation and binding poses for proteins such as PH domains and synaptotagmins. The refined interaction matrix allows for more realistic lipid–protein docking and electrostatic screening, making it particularly suitable for studies of peripheral membrane protein behavior. While MARTINI 2.3P offers enhanced specificity, it still inherits CG limitations, such as the inability to resolve side-chain rotamers, explicit hydrogen bond geometries, or conformational entropy effects. Nevertheless, it provides a valuable middle ground between standard CG efficiency and greater chemical detail.

### 3.5. SIRAH Force Field

The SIRAH (South American Initiative for a Rapid and Accurate Hamiltonian) force field represents a structurally unbiased CG framework developed to bridge the gap between atomistic detail and computational efficiency. Unlike other CG models that often compromise structural fidelity for speed, SIRAH was specifically designed to retain critical biophysical properties, including backbone geometry, electrostatics, and secondary structure definition, particularly for proteins and nucleic acids [75].

SIRAH uses a mapping strategy where the CG beads are placed at the center of geometry or mass of atomistic residues, preserving key chemical moieties and topologies. Its solvent model includes explicit water and ion representations, allowing for long-range electrostatics using particle mesh Ewald (PME), which is often absent in traditional CG approaches. This makes SIRAH particularly suitable for simulating protein–membrane systems, DNA conformations, and peptide anchoring in lipid bilayers under physiological ionic strengths [30,75].

SIRAH’s ability to maintain hydrogen-bond-like interactions and realistic torsional sampling allows for accurate modeling of membrane-bound peptides and protein–lipid interactions, which are often oversimplified in other CG force fields. For instance, its explicit ion support has been shown to capture electrostatically mediated membrane binding events with high fidelity [75]. The force field has also been employed in multiscale simulations, such as LacI–DNA interactions [31], demonstrating its flexibility across system types. Additionally, SIRAH includes secondary structure assignment tools (sirah_ss) and visualization plugins (sirah_vmdtk.tcl) compatible with VMD, enabling frame-by-frame analysis of protein conformations directly from CG trajectories. Backmapping capabilities (sirah_backmap) further enhance its utility by allowing conversion of CG trajectories into atomistic representations for structural refinement, enabling hybrid modeling workflows. These features have proven especially useful in lipid–protein interface studies where near-atomistic insight is desired without the high cost of full all-atom simulations [31]. Despite its strengths, SIRAH, like all CG models, suffers from limited conformational entropy and a reduced ability to capture fine-grained side-chain dynamics or water structuring effects near hydrophilic interfaces. However, recent developments have expanded its chemical space and system compatibility, such as support for glycans and advanced backmapping procedures [30]. The continual evolution of SIRAH through the integration of new chemical components, better mapping libraries, and automation tools positions it as a powerful CG force field for biophysical simulations of complex, multiscale systems, including liposomes, viral capsids, and membrane proteins.

### 3.6. ELBA Force Field: Electrostatics-Driven CG Membrane Modeling

The ELBA (electrostatic-based CG) force field presents a fundamentally different philosophy in CG-MD by incorporating explicit electrostatic interactions and directional polarizability into the modeling framework. Originally introduced by Orsi and Essex [32], ELBA bridges the gap between traditional coarse-grained models and atomistic accuracy by representing water as a single-bead dipolar entity, capable of mimicking key features of hydrogen bonding and solvent polarization. Unlike MARTINI, which uses implicit solvation and short-range Lennard-Jones interactions, ELBA incorporates long-range Coulombic forces via Ewald summation, preserving the directional nature of polar interactions. This unique formulation enables a more physically realistic treatment of electrostatic phenomena, making ELBA particularly well-suited for membrane systems characterized by high surface charge, ion flux, or electric field gradients.

One of the hallmark features of ELBA is its dipolar water model, which reproduces essential dielectric and thermodynamic properties of bulk water while facilitating solvent-mediated interactions with charged lipids and ions. These water beads possess intrinsic dipole moments, enabling the formation of structured solvation shells around lipid headgroups and ions. As a result, ELBA accurately captures complex interfacial phenomena such as double-layer formation, lipid headgroup reorientation, and membrane polarization dynamics. The strength of ELBA lies in its ability to simulate membrane processes in which electrostatics and polar solvation play central roles. For example, ELBA reproduces key structural properties of lipid bilayers, including APL and bilayer thickness, with high accuracy and consistency with experimental data such as small-angle neutron scattering and X-ray diffraction. It also offers reliable estimates for mechanical metrics like surface tension and compressibility modulus, both of which are crucial for evaluating bilayer elasticity and mechanical response under physiological conditions. Importantly, ELBA captures ion–lipid interactions with high fidelity. For instance, calcium binding to negatively charged lipid headgroups, a critical process in signaling and membrane fusion, is realistically modeled due to the explicit treatment of electrostatics.

Further benchmarks show that ELBA accurately resolves ion distribution profiles across membranes, including electric double layer formation, Debye screening, and counterion condensation. It can also model electrostatic asymmetry across bilayers, which plays a fundamental role in driving spontaneous membrane curvature and enabling processes such as vesicle budding, fusion, or electroporation. By capturing polar solvation effects with dipolar water beads, ELBA also permits detailed simulation of hydration forces, nanopore sensing, and pore formation phenomena that are highly sensitive to dielectric properties and solvent orientation.

Validation studies [32] have shown that ELBA performs well against both atomistic simulations and experimental observables, particularly in systems dominated by electrostatics. In a comparative study of oxidized lipid membranes, Siani et al. [33] found that ELBA more faithfully reproduces membrane disruption under oxidative stress, water penetration, and lipid reorganization than MARTINI, which lacks directional electrostatics. These advantages are particularly valuable in charged bilayer systems, where ELBA accurately models salt-induced curvature, asymmetric lipid distribution, and the development of TM potential gradients. Despite its strengths, ELBA does have limitations. Its explicit electrostatics, computed using Ewald summation or particle–particle–particle–mesh (PPPM) algorithms, leads to increased computational costs compared to simpler CG models. Additionally, ELBA currently lacks fully developed parameters for proteins, glycolipids, and other membrane components, which restricts its direct application to multi-component biological systems without further reparameterization.

In conclusion, ELBA expands the capabilities of coarse-grained membrane modeling by introducing directional electrostatics and dipolar solvent models, providing a unique tool for investigating systems where charge interactions, polar solvation, and electric field gradients are mechanistically critical. It complements force fields like MARTINI by offering improved realism in electrostatic-driven processes such as ion translocation, membrane fusion, curvature formation, and nanopore transport phenomena.

### 3.7. SDK and SPICA Force Fields: CG Models for Lipid–Protein Systems

The SDK (Shinoda–DeVane–Klein) and SPICA (surface property-interpolated coarse-grained approach) force fields are intimately related and stem from the same development framework. SPICA is an evolution of SDK, aimed at enhancing the physical realism and range of biomolecular systems modeled using CG simulations. SDK was originally developed as a bottom-up CG force field using force-matching techniques to reproduce atomistic thermodynamic properties such as density, compressibility, and surface tension. It employs an implicit solvent representation and was successfully applied to simulate lipid membranes, surfactants, and self-assembling systems [76]. Recognizing the limitations of the SDK in representing structural membrane properties and membrane protein interactions, SPICA was introduced as its successor. While SDK laid the foundation, SPICA introduced refinements to improve accuracy, especially in lipid-protein systems:**Systematic reparameterization:** SPICA retains the original SDK structure but reparameterizes nonbonded interactions using surface property interpolation to match experimental and atomistic data for APL, bilayer thickness, bending rigidity, and line tension [28].**Compatibility with proteins:** SPICA introduces CG models for amino acids, allowing simulations of protein insertion, dimerization, and conformational dynamics in lipid environments. These models were designed to be fully compatible with SPICA lipids [29].**Improved physical fidelity:** SPICA reproduces depth penetration profiles and tilt angles of transmembrane helices with high accuracy, outperforming previous CG models and closely matching experimental data.

Rather than treating SPICA and SDK as separate models, they should be understood as sequential iterations of the same CG framework. The SPICA force field builds upon SDK by systematically enhancing its ability to represent complex membrane systems, especially those involving proteins and heterogeneous lipid compositions.

In practical applications, SPICA/SDK has been employed to simulate a broad range of biophysical phenomena:**Cholesterol-induced domain formation:** SPICA captures cholesterol-driven phase separation and lateral domain formation in mixed bilayers, consistent with experimental observations of liquid-ordered and liquid-disordered phases [28].**Membrane protein dimerization:** The force field successfully models protein–protein association in bilayers, enabling the prediction of binding orientations and free energy landscapes.**Vesicle curvature and stress:** SPICA allows the study of large, curved membrane structures, accounting for curvature-induced stress, bending modulus, and elastic deformation.**Partitioning of amphiphiles:** The CG framework reproduces preferential localization and partitioning behavior of amphiphilic and hydrophobic small molecules in bilayers, important for drug delivery and sensor design.

By bridging the strengths of bottom-up parameterization with top-down experimental validation, SPICA represents a mature, robust CG model for simulating realistic biological membranes. It is especially valuable in applications where both membrane mechanics and protein interactions must be accurately resolved.

### 3.8. ML-Based CG Force Fields

Recent progress in CG-MD has been significantly shaped by the incorporation of ML techniques, which enable data-driven development of interaction potentials derived directly from atomistic simulations. Among the most notable frameworks is CGNet, a neural network-based CG model introduced by Wang et al. [34], which utilizes a variational force-matching principle to train flexible potential energy functions. This approach allows the model to reproduce atomistic-level forces at a coarse-grained resolution by minimizing the difference between forces observed in high-resolution simulations and those predicted by the learned CG potential.

Unlike traditional CG force fields, which depend on predefined interaction forms and are limited by their pairwise or isotropic nature, ML-based models such as CGNet offer a non-parametric alternative that can capture complex, nonlinear, and many-body interactions without relying on restrictive functional assumptions. In CGNet, deep neural networks are employed to represent the potential of mean force as a function of CG coordinates, thereby enabling the modeling of intricate molecular behavior that is difficult to encode through analytic expressions alone. These models have demonstrated strong performance in capturing high-dimensional free energy surfaces, as well as dynamic behaviors that align with reference atomistic simulations.

A key strength of these data-driven methods lies in their adaptability and generalizability. Once trained, CGNet models can be transferred across thermodynamic conditions and molecular environments with minimal loss in accuracy, provided the training set adequately samples the relevant configurational space. Additionally, the automatic parametrization pipeline offered by machine learning reduces the human effort and subjectivity inherent in traditional CG force field construction, enabling a more scalable and systematic workflow.

Beyond CGNet, recent studies have expanded the application of ML-based CG force fields to a variety of chemical and biological systems, including polymers, organic molecules, and heterogeneous biomembranes. For example, Ye et al. [36] explored a range of machine learning architectures, including kernel methods, graph neural networks, and variational autoencoders, to construct CG models that effectively capture thermodynamic, structural, and dynamic properties across multiple scales. These models have been applied to simulate complex phenomena such as membrane protein interactions, lipid phase behavior, and molecular self-assembly processes.

Despite their promise, challenges remain regarding the interpretability of ML-derived potentials, the need for large and representative training datasets, and the robustness of model generalization across chemical diversity. Nevertheless, the integration of machine learning into CG model development represents a paradigm shift in the field, offering unprecedented accuracy and flexibility for studying soft matter and biomolecular systems.

Together, these modeling frameworks and simulation observables form the basis for computational exploration of liposome design, behavior, and function in drug delivery contexts.

## 4. Liposome Self-Assembly and Functionalization Strategies

This section reviews (I) the mechanisms of liposome self-assembly captured by CG-MD, (II) the biophysical implications of PEGylation, and (III) alternative surface functionalization approaches, including peptides, zwitterionic polymers, and antibodies.

### 4.1. Self-Assembly via CG-MD

Beyond spontaneous formation, liposome surfaces are often functionalized to modulate circulation time and targeting. CG-MD simulations have become indispensable in elucidating the mechanisms of liposome formation. These processes are often challenging to capture experimentally or through all-atom simulations due to computational constraints. By simplifying molecular representation, CG models such as MARTINI facilitate the spontaneous self-assembly of lipid aggregates, including micelles, bilayers, and vesicles, from disordered mixtures of lipids and solvents.

The MARTINI force field enables simulations of large systems over extended timescales (microseconds to milliseconds), allowing the observation of vesicle formation without external templates. The self-assembly process is governed by a delicate interplay between enthalpic and entropic contributions, modulated by lipid geometry, saturation, temperature, hydration, and additive effects.

Wang et al. [5] investigated passive drug loading in vesicular systems using coarse-grained molecular dynamics CG-MD simulations. Their study examined how protonated and neutral drug molecules interact with different membrane topologies, including unilamellar vesicles and open bilayer structures. Rather than simulating vesicle formation from dispersed lipids, the authors analyzed pre-assembled vesicles under equilibration to explore drug partitioning behavior across lipid environments. As shown in Figure 6, the system evolves to reveal how drug molecules associate with membranes of varying curvature and structural organization.

Parchekani et al. [48] examined how vitamin C affects DOPC/DOPE bilayers. Their simulations, starting from randomized lipid distributions, showed that vitamin C induces membrane curvature and promotes vesicle formation. Over hundreds of nanoseconds, lipids self-organized into closed vesicles (Figure 7 and Figure 8).

Duran et al. [61] emphasized the influence of hydration on lipid assembly. Their simulations (Figure 9) spanned from 0 to 1850 ns and captured the transition from dispersed lipids to fully enclosed vesicles. Omitting water and ethanol beads clarified lipid dynamics.

Lee et al. [60] studied PEGylated lipid assemblies and found that increasing PEG content altered membrane morphology by introducing steric hindrance. High PEG concentrations promoted the transition from sheets to vesicles via wormlike micelles (Figure 10).

These studies underscore key molecular factors influencing vesicle self-assembly:**Lipid molecular geometry and unsaturation:** The shape and flexibility of lipid molecules strongly dictate the spontaneous curvature and stability of bilayer assemblies. Conical lipids such as DOPE and DOPC possess unsaturated tails that introduce kinks, reducing tail packing efficiency and encouraging membrane curvature. DOPC, for instance, has a cis-double bond in both acyl chains, making it favorable for flexible bilayer and vesicle formation. In contrast, cylindrical and saturated lipids like DPPC promote tightly packed, less curved, and more ordered membranes due to their straight chains and strong van der Waals interactions. CG-MD studies have shown that mixed lipid systems with unsaturated components are more likely to undergo curvature transitions and vesicle formation [5,48,60].**Hydration and temperature:** Hydration is essential for lipid diffusion and bilayer flexibility. Water molecules solvate lipid headgroups, decreasing interfacial tension and supporting headgroup repulsion, both of which are critical for spontaneous self-assembly. In CG simulations, inadequate hydration can lead to artifactually stable planar structures or incomplete vesicle closure. Temperature also plays a vital role by modulating lipid fluidity. Elevated temperatures increase lateral diffusion, reduce bilayer viscosity, and enhance entropic forces, collectively lowering the energy barrier for vesicle closure and fusion. For example, Wang et al. [5] observed that lipid mobility and vesicle deformation were significantly enhanced at higher temperatures, especially for unsaturated lipids.**Additive and solute effects:** The presence of small molecules or macromolecular additives can significantly alter bilayer behavior. Vitamin C, as shown by Parchekani et al. [48], disrupts planar lipid arrangements and facilitates membrane curvature by locally reducing order and introducing positive curvature stress. Polyethylene glycol (PEG), a polymer frequently used in stealth liposomes, introduces steric hindrance between adjacent lipid headgroups and increases osmotic repulsion. This steric crowding leads to bending instabilities and promotes the transition from lamellar to vesicular structures, as demonstrated in Lee et al. [60]. These additives are thus key tools in tuning membrane properties in both simulations and experimental formulations.**Topological transitions and intermediate states:** The self-assembly of liposomes involves a series of transient intermediate morphologies such as micellar clusters, wormlike tubules, budding structures, and open bilayer sheets. CG-MD is particularly adept at capturing these topological transitions due to its ability to simulate microsecond timescales and large system sizes. Events such as pore formation, vesicle fusion, and neck closure can be visualized in detail, offering mechanistic insight into the dynamic pathways of vesicle formation and remodeling [60,61]. These events are driven by curvature stress and entropic minimization, and are often influenced by system composition and temperature.**System size and simulation conditions:** The spatial dimensions and initial configuration of the CG simulation box greatly affect the vesicle assembly trajectory. A high lipid-to-solvent ratio can favor micelle formation over vesicle closure, while inadequate box size may artificially restrict curvature. The symmetry between inner and outer leaflet composition also determines spontaneous curvature, especially in large vesicles. Moreover, entropy-driven fluctuations, lipid flip-flop, and curvature-induced stress become more prominent in larger systems, as seen in Duran et al. [61]. Thus, careful selection of system parameters is essential for producing realistic vesicle morphologies in silico.

Together, these insights demonstrate how CG-MD, particularly with the MARTINI force field, enables realistic modeling of liposome formation under physiological and experimental conditions. The mechanistic understanding gained from such simulations is essential for the rational design of liposomal carriers in drug delivery, vaccine formulation, and synthetic biology.

### 4.2. PEGylation and Stealth Behavior

While PEGylation remains widely used, several alternatives are gaining prominence due to their enhanced selectivity and reduced immunogenicity. PEG functionalization, or PEGylation, is a well-established strategy to improve the biocompatibility, systemic retention, and therapeutic performance of liposomal drug carriers. By covalently grafting PEG chains onto the lipid bilayer surface, liposomes acquire a protective hydrophilic corona that imparts a “stealth” characteristic, minimizing recognition by the mononuclear phagocyte system (MPS) and preventing opsonin adsorption. This modification significantly prolongs circulation time and enhances passive targeting efficiency via the enhanced permeability and retention (EPR) effect.

CG-MD simulations have provided valuable insights into the structural and dynamic consequences of PEGylation on lipid membranes. PEG chains exhibit two primary conformational regimes: the “mushroom” regime (low grafting density), where PEG chains form compact coils with minimal overlap, and the “brush” regime (high grafting density), where PEG chains stretch outward due to mutual repulsion. Lee et al. [60] systematically characterized these regimes using DPPE-PEG45 systems, showing that increased PEG density induces progressive morphological transitions from planar lamellae to wormlike micelles and closed vesicles.

The study by Woo and Lee [77] demonstrated how PEGylation alters lipid packing by introducing steric hindrance and hydration repulsion forces that destabilize flat bilayers and promote curvature (Figure 11). These effects result in spontaneous self-assembly into vesicular structures, particularly in PEG-rich systems. Their CG-MD simulations highlight the critical role of PEG density in triggering membrane remodeling through curvature stress.

Lemaalem et al. [17] further expanded this understanding by examining how PEGylation affects the mechanical properties of the lipid bilayer. Their simulations showed that PEGylated membranes possess increased bending rigidity and reduced lateral diffusion, as illustrated by mean-field interaction potentials (Figure 12). These changes impact vesicle deformability and stability under physiological shear and flow conditions, critical for drug delivery applications.

Experimental validation of PEGylation effects was provided by Jokerst et al. [12], who demonstrated that PEGylated liposomes evade macrophage uptake and exhibit prolonged blood circulation. Their schematic illustrates PEG’s ability to inhibit opsonin binding, thereby reducing clearance and enhancing tumor accumulation.

Together, these findings elucidate the multifunctional roles of PEG in modulating liposome behavior:**Steric stabilization:** PEG chains generate a dense, hydrophilic polymer shell that effectively excludes proteins and cellular components from accessing the lipid surface. This steric repulsion prevents nonspecific adsorption (opsonization) and minimizes clearance by the immune system, which is essential for extended circulation times and efficient delivery to target tissues [12].**Membrane mechanics regulation:** PEGylation increases the bilayer bending modulus and modulates the mechanical resilience of liposomes under shear stress. The presence of PEG chains introduces entropic elasticity and increases local viscosity, which collectively reduce lipid lateral mobility and contribute to sustained drug retention and controlled release profiles [17].**Induction of curvature and self-assembly:** At high grafting densities, PEG chains create repulsive steric pressure that disrupts planar packing. This excluded volume effect promotes the spontaneous formation of curved membrane morphologies, such as spherical micelles or vesicles. Lee et al. [60] showed that this curvature induction is a key mechanism for PEG-driven membrane remodeling.**Lipid mobility modulation:** The mobility of lipids in PEGylated membranes is reduced due to PEG-induced crowding and increased interfacial viscosity. This reduction in lipid diffusivity influences the spatial organization of membrane domains and can stabilize nanostructured regions such as lipid rafts [17].**Enhanced in vivo performance:** PEGylated vesicles demonstrate significant improvements in pharmacokinetic profiles, including prolonged circulation half-life, reduced hepatic and renal clearance, and enhanced tumor penetration. These benefits are attributed to PEG’s ability to shield liposomes from immune surveillance and promote passive targeting through the EPR effect [12].

These CG-MD and experimental insights support the rational design of PEGylated liposomes. By tuning PEG chain length, grafting density, and lipid anchor chemistry, researchers can precisely modulate membrane mechanics, curvature, and pharmacokinetics to optimize liposome function for diverse biomedical applications.

### 4.3. Surface Functionalization Beyond PEG

While PEGylation has been extensively utilized to endow liposomes with stealth properties, emerging evidence has highlighted its limitations, such as accelerated blood clearance upon repeated administration and potential immunogenicity. These concerns have spurred the development of alternative surface functionalization strategies, including peptides, antibodies, glycans, and zwitterionic polymers that offer enhanced targeting, prolonged circulation, and reduced immune recognition. MD simulations have played a critical role in exploring the mechanistic basis and performance of these advanced coatings.

#### 4.3.1. Peptide-Based Targeting

Peptides provide high specificity and affinity for overexpressed receptors in tumor microenvironments, making them ideal candidates for active liposomal targeting. Cyclic RGD (Arg-Gly-Asp) peptides, for example, bind with high affinity to integrin receptors such as $\alpha_{v}\beta_{3}$ on angiogenic endothelial cells and tumor tissues. Dubey et al. [78] functionalized liposomes with cyclic RGD peptides and demonstrated significantly enhanced uptake by tumor cells in vitro and in vivo, showcasing the power of receptor-mediated endocytosis.

Amin et al. [79] employed cyclic RGD peptides in dual-functionalized liposomes, enabling both passive targeting via the EPR effect and active binding to tumor vasculature. Their study illustrated how peptide density and surface orientation modulate targeting efficiency, emphasizing the importance of molecular-level tuning.

Recent CG and AA MD simulations have revealed how peptide ligands alter membrane surface dynamics and impact liposome cell membrane interactions. Aronson et al. [80] reviewed simulation-based design of peptide-functionalized liposomes, highlighting insights into linker flexibility, ligand clustering, and lipid–peptide compatibility that govern targeting selectivity and fusion efficiency.

#### 4.3.2. Zwitterionic Coatings

Zwitterionic polymers have emerged as promising alternatives to PEG due to their ultra-low fouling properties, biocompatibility, and charge neutrality. These materials mimic cell membrane phospholipids and form dense hydration shells, effectively repelling proteins and minimizing immune recognition. Li et al. [81] developed zwitterionic lipid nanoparticles (ZwiLNPs) for siRNA delivery, achieving selective organ targeting and reduced off-target accumulation. Their study emphasized how zwitterionic surfaces prevent nonspecific protein adsorption while enabling efficient endosomal escape, both key parameters in therapeutic delivery.

MD simulations have shown that zwitterionic lipid headgroups form stable hydration shells, limit lateral diffusion, and preserve membrane structure. Zhao et al. [82] demonstrated, through simulations and experiments, that pH-sensitive charge self-transformation enables ZwiLNPs to adapt to biological environments, improving membrane interaction, uptake, and siRNA transfection for cholesterol-lowering therapy. Figure 13 summarizes the composition and delivery mechanism of ZwiLNPs, showing their long circulation, targeted tissue accumulation, endosomal escape, and siPCSK9 release to treat hypercholesterolemia.

#### 4.3.3. Antibody and Glycan Functionalization

Antibodies and glycan structures offer high molecular specificity for disease markers, enabling precision delivery. Although more structurally complex, their interactions with membranes can also be explored via MD approaches.

Wei et al. [83] functionalized liposomes with glioma-targeting peptides and observed enhanced transport across the blood–brain barrier. While primarily experimental, their findings motivate further simulations that explore antibody–lipid anchoring, steric repulsion, and ligand accessibility under dynamic conditions. Together, these strategies represent a new frontier in liposomal surface engineering. By incorporating MD simulation data with empirical findings, researchers can rationally design liposomes with improved targeting, immune evasion, and drug release characteristics, expanding their clinical versatility beyond conventional PEGylated systems.

## 5. Drug–Lipid Interactions and Translational Insights

Understanding drug–lipid interactions is crucial for designing liposomal formulations with optimized stability, controlled release, and targeted delivery. These interactions influence the pharmacokinetics, biodistribution, and efficacy of encapsulated therapeutics, thereby directly impacting clinical performance and translational success.

### 5.1. Drug–Lipid Interactions

MD simulations spanning all-atom CG approaches have become essential in elucidating how lipids interact with and encapsulate drugs. These simulations offer mechanistic insights into the roles of membrane curvature, lipid composition, and drug physicochemical properties in modulating encapsulation efficiency, membrane stability, and controlled release.

Zhu et al. [7] employed atomistic MD to investigate liposomes embedded with octanoylated hyaluronic acid. Their results revealed improved bilayer stability and drug retention, facilitated by hydrophobic anchoring and hydrogen bonding between octanoyl groups and lipid tails. These interactions led to denser lipid packing and decreased permeability, underscoring how chemical modifications can enhance structural integrity.

Hudiyanti et al. [19] simulated vitamin C encapsulation in DOPC bilayers with varying cholesterol levels. Their work revealed that moderate cholesterol concentrations (20–30%) optimized bilayer fluidity, enhancing encapsulation and reducing water ingress. In contrast, excessive cholesterol disrupted lipid order and triggered phase separation, ultimately destabilizing the bilayer.

Using CG-MD simulations, Barreto-Ojeda et al. [84] investigated doxorubicin (DOX) interactions with lipid membranes. DOX exhibited a progressive insertion into the bilayer, initially associating with the headgroup region and later penetrating the hydrophobic core. Penetration depth and stability were influenced by phosphatidylethanolamine content and curvature, echoing earlier findings by Siani et al. [38] regarding curvature-dependent DOX localization.

Atomistic simulations by Shirazi-Fard et al. [63] further explored the embedding of hydrophobic drugs near the bilayer midplane. Their study revealed drug-induced local ordering of lipid tails, increased bilayer thickness, and enhanced bending modulus properties that reinforce membrane rigidity under stress, such as shear or dilution.

Dutta et al. [85] examined paclitaxel–lipid interactions, highlighting how drug intercalation creates transient membrane disruptions that facilitate passive drug release. These dynamic defects promote lateral diffusion and transient permeability, offering insights into potential release mechanisms.

A notable advancement by Pilkington et al. [62] demonstrated a liposome-in-liposome system, wherein both experimental and CG-MD techniques were used to model multi-compartmental encapsulation and sequential release. A key design feature of this system was the use of PEGylated lipids to modulate the intermembrane distance between the nested vesicles. As shown in Figure 14, increasing the PEG chain length led to a corresponding increase in bilayer spacing, which improved inner compartment stability and enabled structural tunability [62].

In addition to enhanced encapsulation, the system supported temperature-dependent sequential release of cargo. By tuning the phase transition temperatures of the outer DPPC and inner DSPC membranes, the authors achieved thermally triggered, multi-stage drug release, an effect visualized in Figure 15 [62].

These findings are intricately linked to other parts of this review. PEGylation and surface modifications influence interfacial hydration and steric hindrance, directly modulating drug partitioning, while APL and cholesterol-dependent membrane packing alter bilayer rigidity and tail ordering, thus regulating drug diffusion barriers. Membrane Curvature and Geometric Confinement Effects further affect bilayer asymmetry, drug localization, and release kinetics.

Taken together, MD simulations offer a powerful framework for decoding drug–lipid interactions across scales. By correlating atomic-level behaviors with macroscopic performance indicators such as encapsulation efficiency, release timing, and mechanical stability, MD approaches provide critical insights for the rational engineering of liposomal therapeutics.

Variations in phospholipid tail saturation, headgroup identity, and cholesterol content critically determine the biophysical behavior of lipid bilayers. These compositional parameters influence bilayer fluidity, lateral diffusion, bending rigidity, membrane thickness, and spontaneous curvature. CG-MD simulations offer a robust platform to systematically study these effects at both the molecular and mesoscopic scales. They provide insights into how specific lipid compositions modulate liposome morphology, curvature stress, and vesicle stability across a range of biologically relevant conditions [23,24,25,59].

### 5.2. Case Studies: Marketed Liposomal Drugs via CG-MD

Several FDA-approved liposomal drugs such as Doxil^®^, AmBisome^®^, and Onivyde^®^ have been extensively studied using CG and atomistic MD simulations. These studies provide mechanistic insight into how formulation parameters such as PEGylation, lipid composition, and cholesterol content influence vesicle morphology, drug retention, and delivery efficiency.

Magarkar et al. [18] investigated the interaction of amphotericin B (AmB) with liposomal membranes, modeling the FDA-approved formulation AmBisome^®^. Their simulations revealed that the inclusion of cholesterol at high concentrations (above 40 mol%) results in a tightly packed bilayer that reduces AmB aggregation and minimizes membrane disruption. As shown in Figure 16, PEG density and cholesterol content modulate ion distributions and membrane surface behavior, which in turn contribute to increased drug retention and circulation stability.

Similarly, Lee et al. [66] used atomistic MD simulations to study Doxil^®^, a PEGylated liposomal formulation of doxorubicin. They demonstrated that PEG chains provide steric stabilization and reduce opsonization, while cholesterol modulates bilayer rigidity and suppresses premature drug leakage. Importantly, doxorubicin was observed to form ordered aggregates within the aqueous core and engage in electrostatic interactions with anionic lipid headgroups.

Mahdavi et al. [86] examined irinotecan-loaded liposomes (Onivyde^®^), focusing on the interaction between the encapsulated drug and lipid headgroups. Coarse-grained simulations revealed that irinotecan forms stable clusters in the aqueous core, while its positively charged groups transiently anchor to phosphate headgroups, resulting in local bilayer thinning. As illustrated in Figure 17, PEGylation length plays a significant role in modulating drug carrier interaction strength, a principle that translates to PEGylated liposomes.

Pilkington et al. [62] further explored how lipid composition and PEGylation influence membrane mechanical properties. Their MD simulations demonstrated that increasing PEG density and cholesterol content enhances membrane bending rigidity, which is an important determinant of vesicle stability during circulation and cellular uptake.

Piasentin et al. [50] complemented these structural insights by employing COSMOmic and MD simulations to predict drug–membrane partition coefficients. Their findings emphasized cholesterol’s regulatory role in drug localization and permeation within complex lipid bilayers, providing a quantitative framework for rational liposome design.

Collectively, these simulation-based case studies validate the design principles discussed in earlier sections. They highlight cholesterol’s dual role in membrane stabilization and selective permeability, PEGylation’s contribution to immune evasion and structural longevity, and the influence of electrostatic and hydrophobic interactions on drug loading and release in clinically relevant formulations.

### 5.3. Limitations and Future Directions

While MD simulations, particularly CG and atomistic approaches, have significantly advanced our understanding of liposome behavior, important challenges remain that must be addressed to achieve truly predictive and translational modeling in drug delivery applications. This section outlines current limitations and future directions that could overcome these barriers and expand the utility of MD in liposomal nanomedicine.

#### 5.3.1. Timescale and Sampling Limitations

One of the most pressing limitations in MD simulations is the restricted timescale they can practically access. Atomistic MD simulations typically capture dynamics on the nanosecond to microsecond range, whereas biologically relevant processes such as membrane fusion, lipid raft formation, vesicle trafficking, drug release, and endocytosis often occur on millisecond to second timescales. Even CG models like MARTINI, while extending the feasible simulation duration by simplifying system resolution, generally remain in the microsecond regime. This gap limits our ability to observe rare but essential events in drug delivery, such as vesicle deformation under shear flow, drug partitioning dynamics within heterogeneous membranes, or lipid flip-flop during uptake. To address this, enhanced sampling techniques such as metadynamics, replica exchange, and temperature-accelerated MD have been adopted. Additionally, Markov state models (MSMs), as detailed by Noé et al. [35], provide a rigorous statistical framework to reconstruct long-timescale kinetics from short trajectories by identifying metastable conformational states and the transitions between them. MSMs have already shown promise in protein folding and membrane fusion studies, and their application to liposome modeling could significantly improve the resolution of slow kinetic processes. Further integration with ML-based reaction coordinate discovery is expected to enhance sampling efficiency even more.

#### 5.3.2. Force Field Limitations and Transferability

The accuracy and predictive power of MD simulations depend heavily on the quality and applicability of the underlying FFs. In atomistic models, FFs such as CHARMM, AMBER, and OPLS-AA capture detailed molecular interactions, but their computational expense limits their use for large vesicular systems. CG FFs like MARTINI, while computationally tractable for large-scale lipid membranes, suffer from approximations that can lead to inaccurate representations of electrostatic interactions, hydration effects, and specific lipid–drug binding modes.

Such limitations become particularly evident when simulating functionalized lipids, charged therapeutic molecules, or stimuli-responsive vesicles, which often exhibit subtle but crucial interaction networks that CG force fields may overlook. Moreover, transferability across different lipid compositions or temperature/pH conditions is often poor, necessitating repeated reparameterization. Recent advances, such as AI-driven force field parameterization, demonstrated by Wang et al. [34], offer an exciting solution. By leveraging large datasets and optimization algorithms, machine learning can be used to generate FF parameters that are chemically specific and dynamically adjustable, improving simulation fidelity across diverse lipid systems. Additionally, hybrid force fields that couple atomistic and CG resolutions (multiscale models) provide a promising compromise, maintaining key interaction details at drug–lipid interfaces while preserving scalability.

#### 5.3.3. Limitations of Idealized Liposome Models in MD Simulations

Most molecular dynamics (MD) simulations of liposomes rely on highly idealized representations that, while computationally tractable, fall short of capturing the complexity of real biological systems. These simulations often model liposomes as symmetric bilayers or vesicles composed solely of a few well-characterized lipids, typically DOPC, DPPC, cholesterol, or PEGylated lipids, immersed in water or saline. In contrast, in vivo liposomes encounter a heterogeneous and dynamic biological environment that includes serum proteins, extracellular matrices, cellular interfaces, and active transport mechanisms.

This biological milieu introduces numerous factors such as protein corona formation, immune recognition, enzymatic degradation, and mechanical deformation by tissue interfaces that profoundly influence liposome stability, biodistribution, and therapeutic efficacy. For example, the adsorption of plasma proteins onto the liposomal surface can obscure targeting ligands, accelerate opsonization, and promote clearance by the mononuclear phagocyte system (MPS), ultimately reducing circulation half-life [87]. Enzymatic activities at the membrane interface may also degrade lipid components or modulate membrane permeability in ways not represented in simplified models.

To bridge this realism gap, emerging strategies include hybrid modeling approaches that combine CG and atomistic representations, multi-scale simulations, and integration of experimental data such as cryo-electron microscopy (cryo-EM), NMR spectroscopy, and small-angle X-ray scattering (SAXS) [23]. In addition, coupling MD simulations with organ-on-chip platforms, microfluidic flow devices, or advanced in vitro systems offers a promising route for validating and calibrating model parameters against physiologically relevant data [2]. Implicit representations of protein corona dynamics, mechanical forces, and interfacial heterogeneity may also be incorporated into MD frameworks to simulate more realistic biological responses. While MD simulations provide powerful insights into liposomal behavior at the molecular level, embracing biological realism through multi-scale modeling, experimental validation, and the incorporation of physiological complexity remains essential for translating simulation results into clinical applications.

#### 5.3.4. Integration of AI and Data-Driven Modeling

The integration of AI and ML into MD simulation pipelines has emerged as a transformative trend. By analyzing large datasets from previous simulations and experiments, ML algorithms can predict key drug delivery metrics such as encapsulation efficiency, release rate, lipid partition coefficients, and membrane permeability with increasing accuracy and speed.

For example, Torres et al. [4] demonstrated the utility of deep learning models trained on lipid–drug interaction data to predict optimal lipid compositions for hydrophobic and amphiphilic drugs. This reduces the need for trial-and-error simulations and accelerates formulation development. Additionally, graph neural networks (GNNs) have shown promise in capturing the spatial relationships between lipid molecules and drug compounds, enabling more interpretable and transferable predictions.

When combined with physics-based simulations, these data-driven methods allow for active learning loops, where AI models guide the selection of simulation parameters or conditions to maximize informational gain. Such hybrid workflows also support the automated generation of FF parameters, real-time simulation steering, and virtual screening of candidate lipid–drug systems, significantly reducing the barrier to entry for non-experts in the field.

#### 5.3.5. Rational Design of Liposomal Drug Delivery Systems

Conventional liposomes have largely relied on passive drug delivery mechanisms. However, recent efforts have shifted toward designing stimuli-responsive and targeted liposomes that react to specific biochemical cues such as pH, temperature, oxidative stress, and enzymatic activity. These systems can significantly enhance site-specific drug release, reduce systemic toxicity, and improve therapeutic efficacy.

Wang et al. [3] comprehensively reviewed multi-stimuli-responsive liposomes, including pH-sensitive carriers that exploit acidic tumor environments, redox-sensitive systems responsive to intracellular glutathione levels, and thermosensitive formulations activated by mild hyperthermia. Incorporating cardiolipin into vesicles for mitochondrial targeting has also shown promise in selectively delivering drugs to cancer cell mitochondria.

MD simulations provide an essential platform for rationally designing and evaluating such smart systems. By modeling the conformational response of lipids or polymers to environmental triggers, simulations can predict how structural transitions influence drug retention, permeability, and membrane fusion. Furthermore, the availability of patient-specific omics data (e.g., lipidomic, proteomic) enables the development of personalized nanomedicine strategies, where liposome behavior is modeled and optimized for individual patients based on their unique biological profile.

## 6. Conclusions

Liposomes are central to numerous biomedical and biotechnological applications, including drug delivery, gene therapy, biosensing, and vaccine delivery. Their tunable size, composition, and surface characteristics make them ideal candidates for encapsulating both hydrophilic and hydrophobic agents. However, unlocking their full potential requires a deep molecular-level understanding of their behavior under physiological and pathological conditions. MD simulations have emerged as a transformative tool in this regard, enabling visualization and quantification of the structural, mechanical, and dynamic properties of liposomes with high resolution.

This review has explored the landscape of MD simulations applied to liposomal systems, with a focus on CG approaches that offer a compelling trade-off between computational efficiency and biophysical realism. While AA simulations provide detailed accuracy, they are often prohibitive for studying large vesicles or long-timescale processes. CG models reduce system complexity while preserving essential physical interactions, thus facilitating simulations of mesoscopic liposome behavior.

Among existing CG force fields, the MARTINI framework is widely used due to its balance between speed and transferability. Its evolving versions (e.g., MARTINI 3) offer expanded chemical coverage and improved accuracy. Nonetheless, limitations such as short range, non-polarizable interactions can affect its reliability in systems with strong electrostatic or polar contributions. Alternatives like ELBA, SIRAH, SDK, and SPICA offer varying solutions: ELBA includes dipolar water and long-range Coulombic interactions; SIRAH achieves near-atomistic resolution; SDK and SPICA employ force-matching and surface property-based approaches to simulate complex lipid–protein and lipid–solvent interactions.

MD simulations have proven essential in quantifying key liposomal parameters such as APL, bilayer thickness, lateral pressure profiles, and compressibility modulus metrics crucial for understanding membrane rigidity, curvature stress, and vesicle stability. For instance, cholesterol concentration alters bilayer organization in a nonlinear fashion, promoting condensation at moderate levels and phase separation at higher concentrations. Similarly, curvature effects in spherical vesicles induce asymmetries between leaflets, influencing packing, tail ordering, and interleaflet dynamics.

Simulations also shed light on dynamic events like lipid flip-flop, vesicle fusion, and interleaflet coupling phenomena that are integral to cargo delivery, membrane remodeling, and signal transduction. Notably, CG simulations reveal that membrane curvature and lipid asymmetry significantly affect these dynamics, particularly in the presence of cholesterol.

Looking ahead, the integration of ML with MD simulations is poised to revolutionize the field. Emerging ML-based force fields, such as CGNet, learn effective potentials directly from atomistic data, offering adaptive and accurate modeling of chemically diverse membrane systems. These tools enable in silico screening of lipid compositions and vesicle designs with an eye toward optimizing clinical performance, such as targeted delivery, reduced toxicity, and improved release profiles. With growing access to patient-derived data, future liposomal formulations may become increasingly personalized, leveraging AI to tailor nanocarriers for specific biological contexts.

Taken together, MD simulations anchored in biophysically accurate force fields and empowered by emerging computational strategies have become indispensable for understanding and engineering liposome-based systems. They serve not only as a bridge between theory and experiment, but also as a predictive engine for advancing nanomedicine, membrane biophysics, and drug delivery technologies.

## Figures and Tables

**Figure 1 membranes-15-00259-f001:**
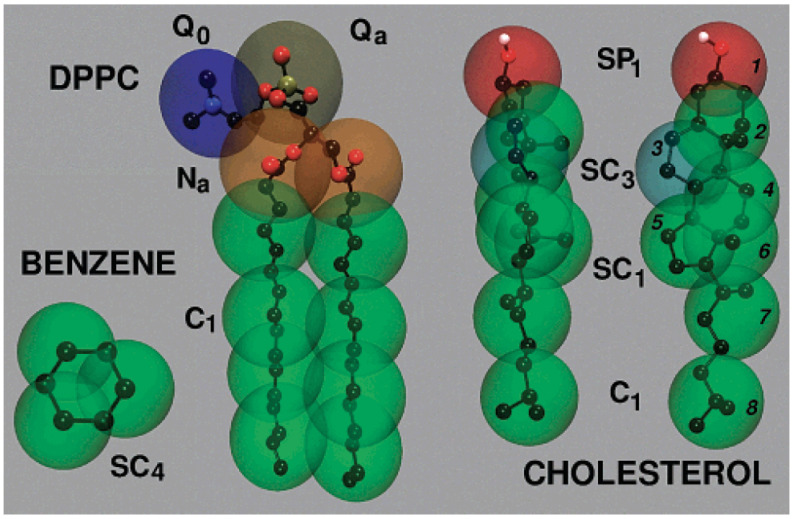
Mapping between the chemical structure and the CG model for DPPC, cholesterol, and benzene. The CG bead types, indicating relative hydrophilicity, are shown. “S” denotes special bead types used to model ring structures. Reproduced from Marrink et al. [23], with permission. ©2007 Elsevier.

**Figure 2 membranes-15-00259-f002:**
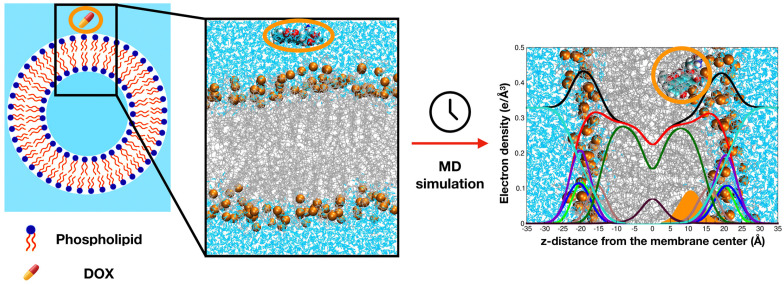
Schematic and simulation-based electron density profile of a doxorubicin-loaded lipid bilayer system. The left panel shows a vesicle composed of phospholipids (blue/red) and DOX (orange), while the center and right panels show atomistic snapshots and electron density distributions of DOX^+^ penetrating a sphingomyelin-based bilayer. Reproduced from Siani et al. [38], with permission. © 2022 Elsevier.

**Figure 3 membranes-15-00259-f003:**
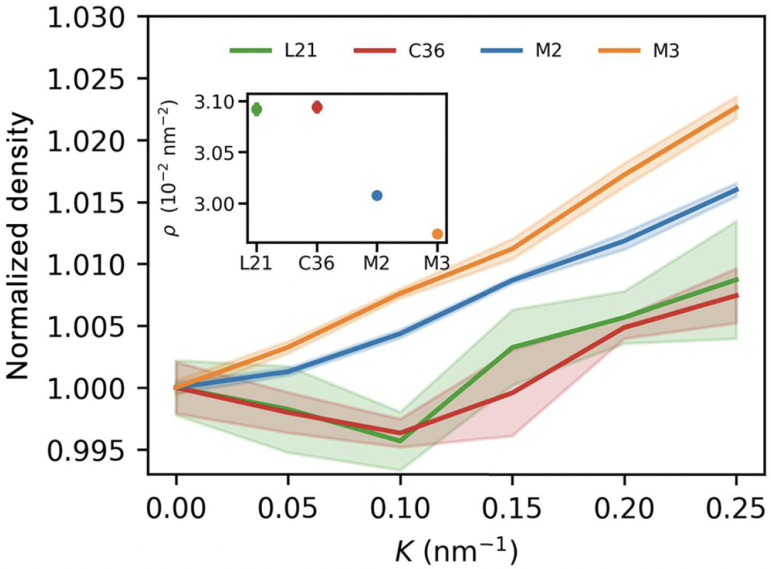
Curvature-dependent POPC bilayer area density. All values were normalized to 1 at K=0 nm^−1^, with curvature leading to local variations in lipid packing density. Reproduced from Domanska et al. [16], with permission. © 2024 Elsevier.

**Figure 4 membranes-15-00259-f004:**
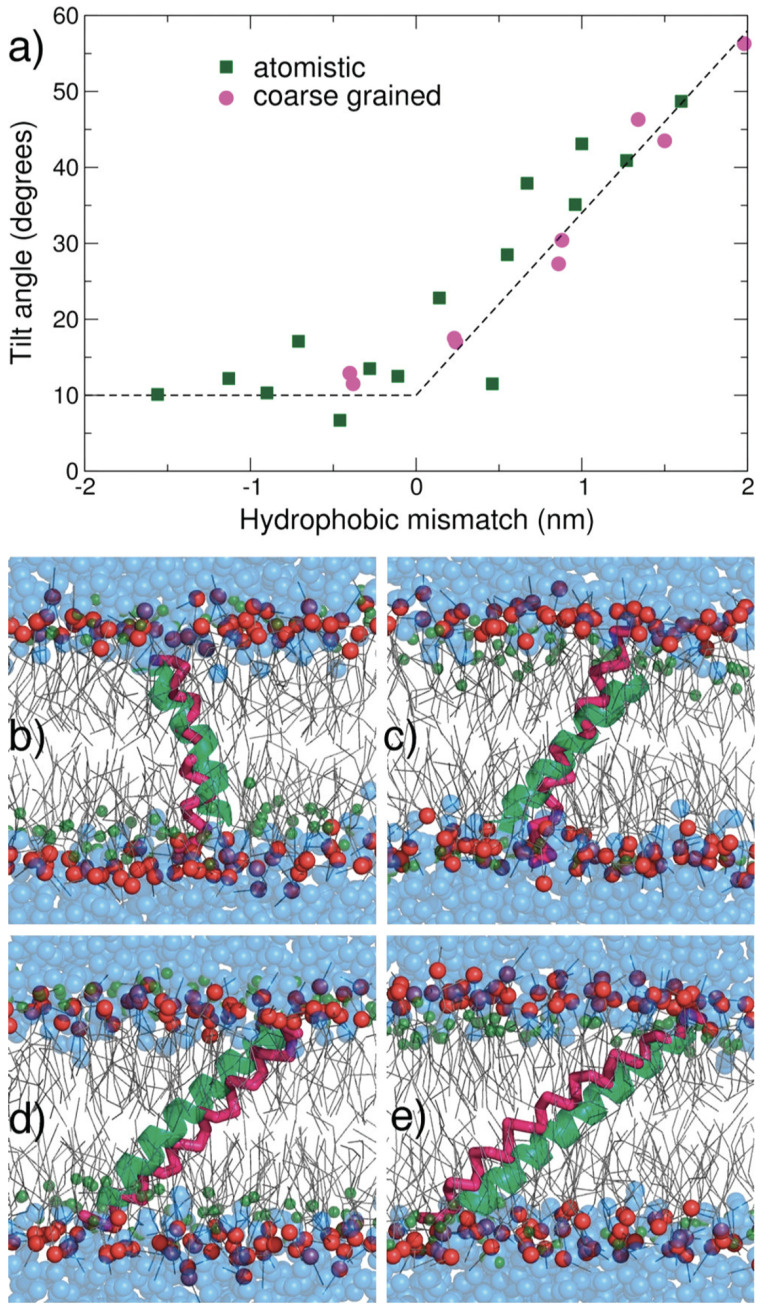
Comparison of CG and atomistic simulations of KALP peptides in DLPC lipid bilayers. (**a**) Tilt angles as a function of hydrophobic mismatch for atomistic (green squares) and CG (pink circles) simulations; (**b**) KALP19 peptide in DLPC lipids; (**c**) KALP23 peptide in DLPC lipids; (**d**) KALP27 peptide in DLPC lipids; (**e**) KALP31 peptide in DLPC lipids. In panels (**b**–**e**), water is shown as blue spheres, lipid phosphate groups as red spheres, peptides as a pink backbone trace, and lipids as grey lines. The peptide from the corresponding atomistic simulation (green helix) is overlaid on the CG peptide for comparison, with phosphorus atoms from atomistic lipids also shown for reference. Reproduced from Monticelli et al. [24], with permission. © 2008 American Chemical Society.

**Figure 5 membranes-15-00259-f005:**
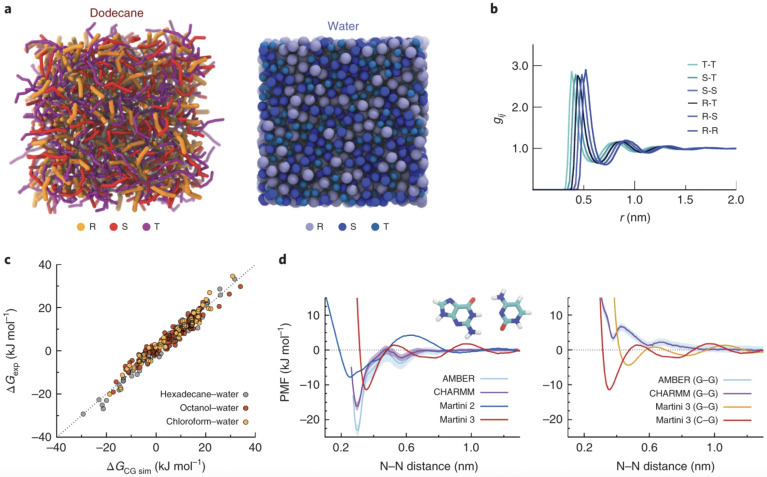
Validation of MARTINI 3 force field: (**a**) simulation snapshots, (**b**) radial distribution functions, (**c**) transfer free energies, and (**d**) hydrogen-bonding PMFs. These enhancements allow for more accurate modeling of lipid–solvent interactions and nucleic acid systems. Reproduced from Souza et al. [25], with permission. © 2021 Springer Nature.

**Figure 6 membranes-15-00259-f006:**
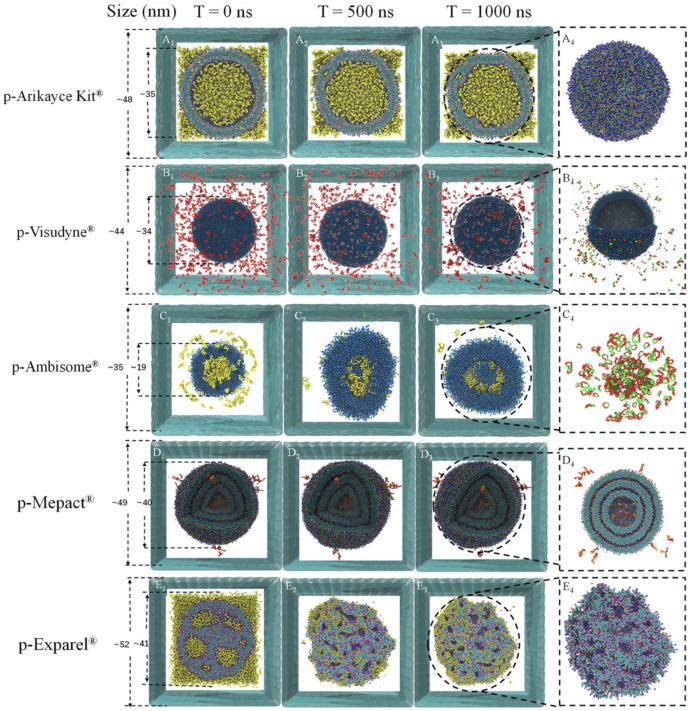
Representative snapshots from CG-MD simulations of passive drug-loading systems with protonated drug molecules at different times (0, 500, and 1000 ns). (A1–A4) p-Arikayce Kit^®^; (B1–B4) p-Visudyne^®^; (C1–C4) p-AmBisome^®^; (D1–D4) p-Mepact^®^; (E1–E4) p-Exparel^®^. These snapshots highlight the equilibration and structural evolution of vesicular systems under different drug–lipid conditions. Reproduced from Wang et al. [5], with permission. © 2025 Elsevier.

**Figure 7 membranes-15-00259-f007:**
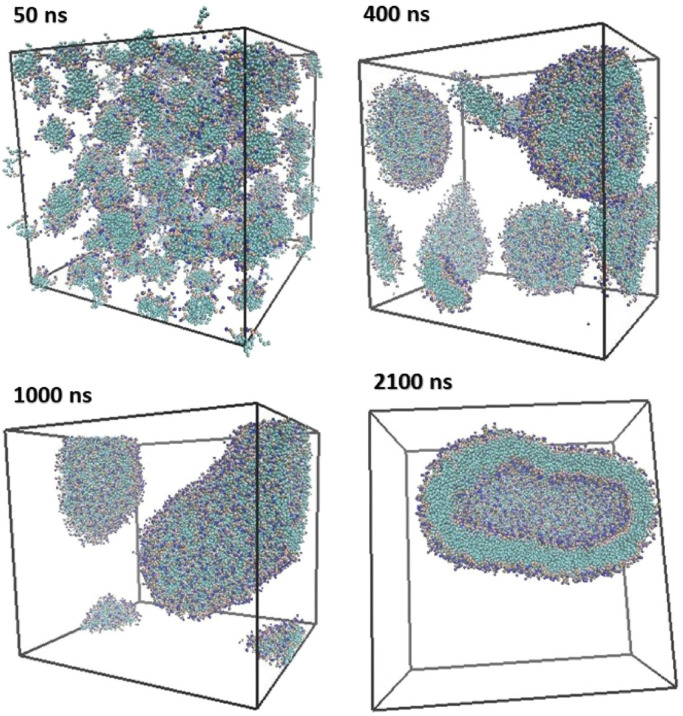
Time-resolved vesicle formation. Lipids assemble into vesicular structures over the course of the simulation. Reproduced from Parchekani et al. [48], with permission. © 2022 Elsevier.

**Figure 8 membranes-15-00259-f008:**
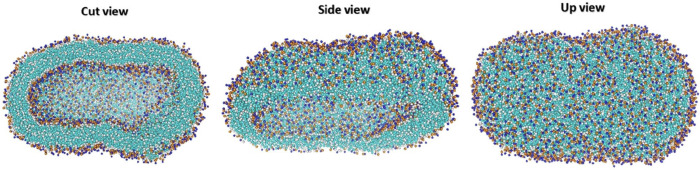
Final vesicle structures shown in multiple views, confirming closure and integrity. Reproduced from Parchekani et al. [48], with permission. © 2022 Elsevier.

**Figure 9 membranes-15-00259-f009:**
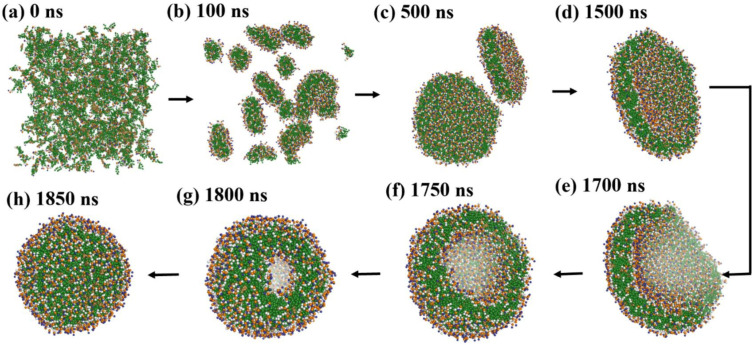
Vesicle formation over time in CG-MD simulations. Lipids coalesce and reorganize into vesicles. Reproduced from Duran et al. [61], with permission. © 2024 Elsevier.

**Figure 10 membranes-15-00259-f010:**
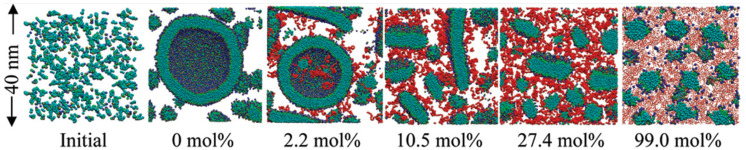
Effects of PEG concentration on vesicle formation. Snapshots at 0 ns and 300 ns for varying DPPE-PEG45 content. Reproduced from Lee et al. [60], with permission. © 2011 Elsevier.

**Figure 11 membranes-15-00259-f011:**
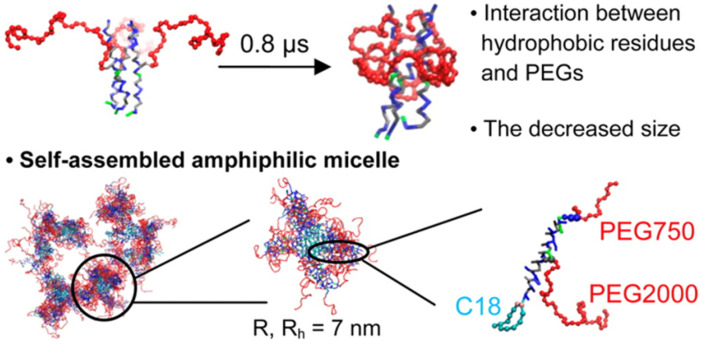
Initial and final snapshots of PEGylated trimeric coiled coils and micelle assemblies. PEG chains demonstrate compact and extended structures influencing self-assembly. Reproduced from Woo et al. [77], with permission. © 2014 Elsevier.

**Figure 12 membranes-15-00259-f012:**
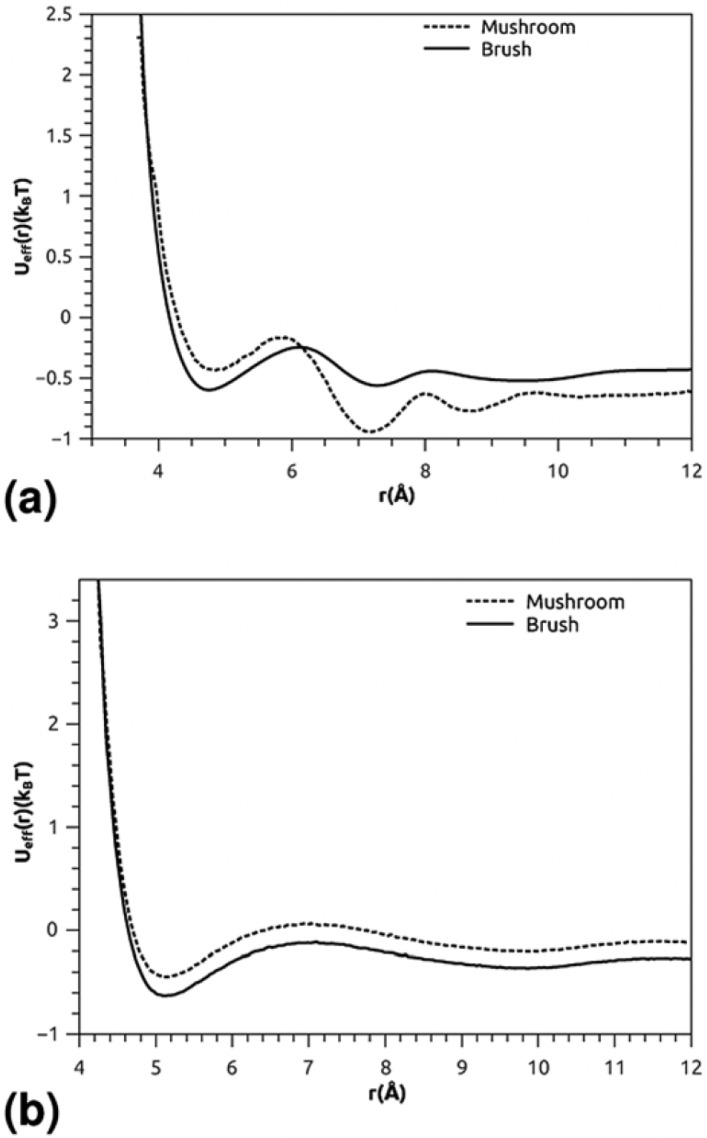
Mean–field interaction potentials ($U_{\mathrm{eff}}\left(r\right)$) for PEGylated lipid components. (**a**) $U_{\mathrm{eff}}\left(r\right)$ of the group of lipopolymer molecules. (**b**) $U_{\mathrm{eff}}\left(r\right)$ of the group of lipid molecules. PEGylation reduces molecular mobility and increases bilayer rigidity. Reproduced from Lemaalem et al. [17], with permission. © 2020 Elsevier.

**Figure 13 membranes-15-00259-f013:**
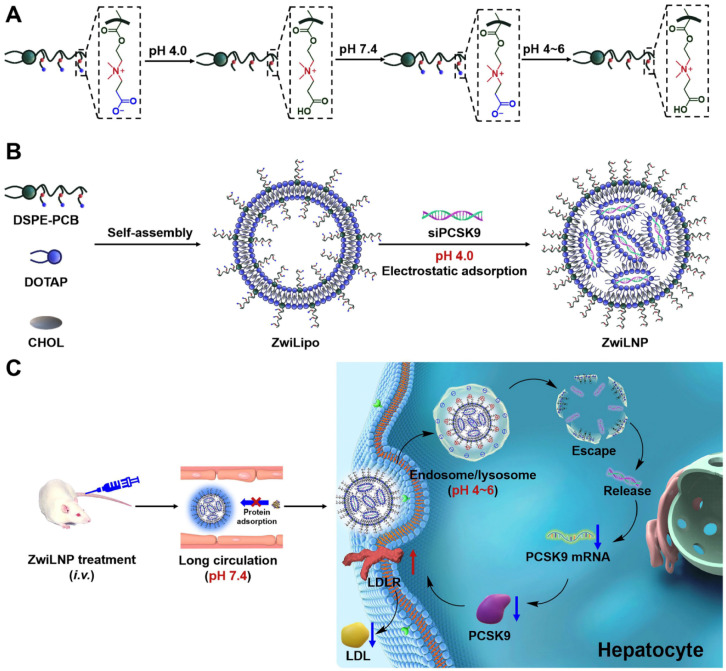
(**A**) Schematic illustration of the pH-sensitive charge self-transformation of zwitterionic polymers used in lipid nanoparticles (ZwiLNPs). (**B**) Composition of ZwiLNPs including zwitterionic polymers, lipids, and siRNA. (**C**) Mechanism of in vivo delivery and therapeutic effect: long-circulating ZwiLNPs accumulate in target tissues, escape endosomes, and release siPCSK9 to treat hypercholesterolemia. Reproduced from Zhao et al. [82], with permission. © 2025 Ivyspring International Publisher.

**Figure 14 membranes-15-00259-f014:**
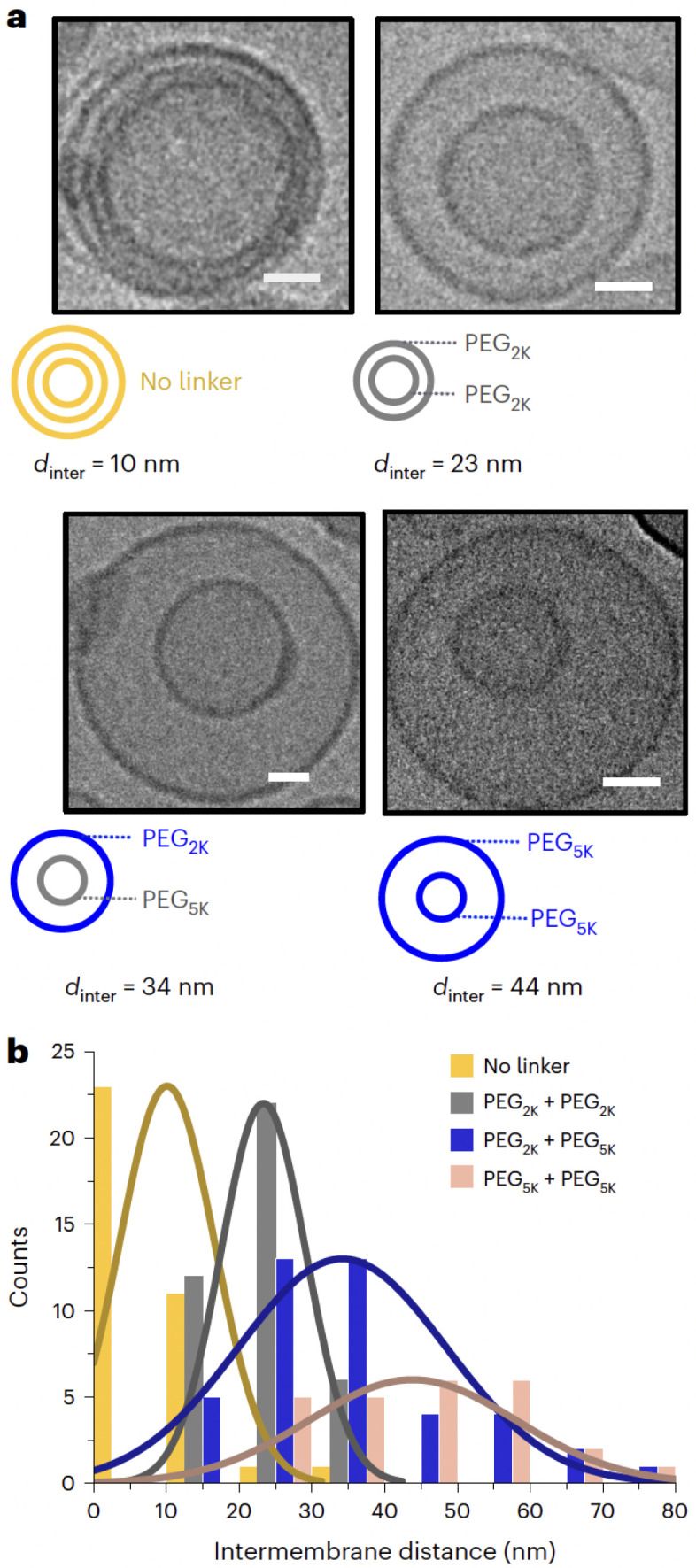
PEG chain length modulates intermembrane distance in liposome-in-liposome systems. Longer PEG linkers increase bilayer spacing, improving compartment stability and encapsulation capacity. (**a**) Cryo-EM micrographs showing $d_{\mathrm{inter}}$ values for different PEG linker combinations. (**b**) Histogram of measured $d_{\mathrm{inter}}$ values with Gaussian fits, demonstrating increased spacing with longer PEG chains. Reproduced from Pilkington et al. [62], with permission. © 2024 Elsevier.

**Figure 15 membranes-15-00259-f015:**
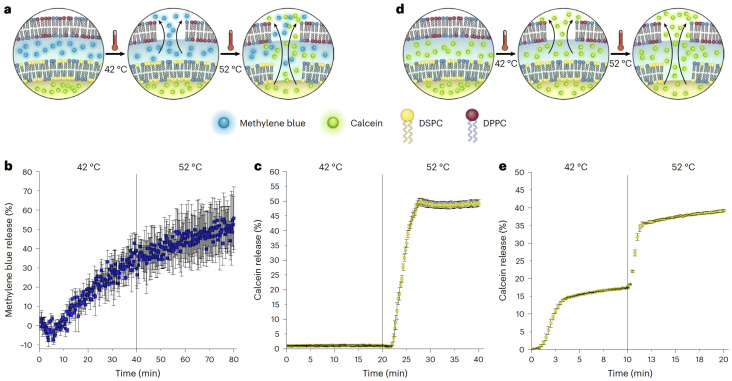
Temperature-triggered sequential release of encapsulated agents from inner and outer bilayers. (**a**) Graphic illustrating multi-stage sequential release of two different cargos (self-quenching dyes) from the two concentrisome compartments. (**b**) Percent release of methylene blue from the intermembrane space, showing release at 42 °C (above the transition temperature of DPPC in the outer bilayer). (**c**) Percent release of calcein from the inner liposome, observed only at 52 °C (above the transition temperature of DSPC in the inner bilayer). (**d**) Graphic showing sequential release of calcein from both compartments in a DPPC/DSPC concentrisome system. (**e**) Percent release of calcein dye, demonstrating two distinct release events corresponding to the transition temperatures of the outer and inner bilayers. Reproduced from Pilkington et al. [62], with permission. © 2024 Elsevier.

**Figure 16 membranes-15-00259-f016:**
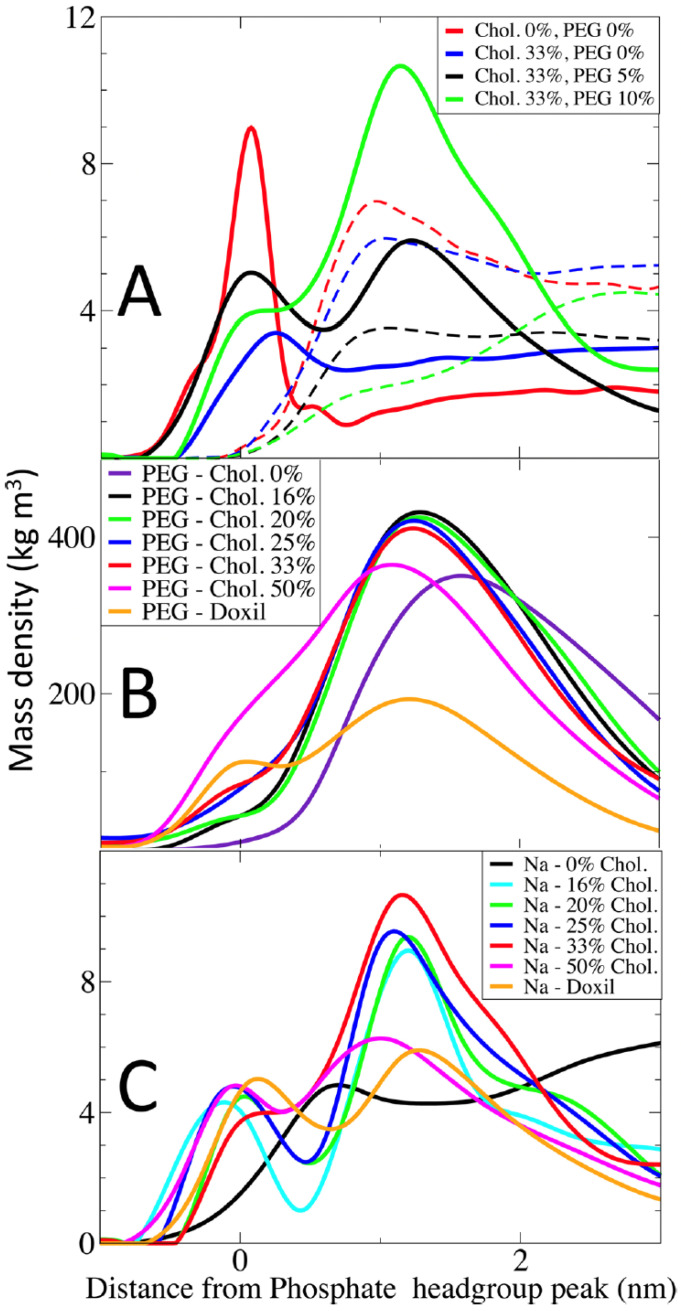
Mass density profiles from CG-MD simulations: (**A**) Na^+^ and Cl^−^ ions at different PEG densities, (**B**) PEG density distribution with varying cholesterol levels, and (**C**) Na^+^ ions affected by cholesterol. These profiles illustrate how PEGylation and cholesterol modulate membrane interface properties and ionic distribution key factors for drug retention and circulation stability. Reproduced from Magarkar et al. [18], with permission. © 2014 Elsevier.

**Figure 17 membranes-15-00259-f017:**
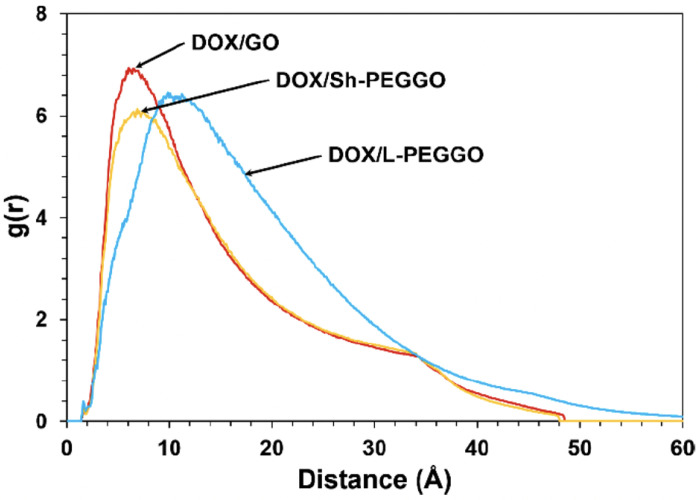
Radial distribution function (RDF) of the center of mass (COM) of doxorubicin (DOX) in different PEGylated graphene oxide nanocarrier systems: DOX/GO, DOX/Sh-PEGGO, and DOX/L-PEGGO. While these systems are not liposomes, the results provide insights into PEG-length-dependent modulation of drug–carrier interactions, which are also relevant to PEGylated liposomes such as Doxil^®^. Reproduced from Mahdavi et al. [86], with permission. © 2020 Elsevier.

**Table 1 membranes-15-00259-t001:** Comparison between AA and CG-MD simulations.

Feature	AA-MD	CG-MD
Resolution	Atomic-level (each atom explicit)	Reduced (typically 3–5 heavy atoms per bead)
Accuracy	High; captures H-bonds and torsions	Moderate; mesoscale properties preserved
Computational cost	High; small time steps; heavy compute	Lower; longer time steps; fewer interaction sites
Typical timescales	ns–µs	µs–ms
Typical system size	Tens–hundreds of nm	Hundreds of nm–µm
Force fields	CHARMM, AMBER, GROMOS, OPLS	MARTINI (v2/v3), SDK, ELBA
Applications	Protein folding; ligand binding; membrane protein dynamics	Membrane remodeling; vesicle dynamics; large assemblies
Advantages	Chemically accurate; high resolution	Efficient; large systems and long timescales
Limitations	Expensive; limited size/timescale	Lower resolution; reduced specificity; needs validation
Integration with experiments	Interpreting NMR, crystallography, spectroscopy	Comparing with microscopy and scattering

## Data Availability

No new data were created or analyzed in this study. Data sharing is not applicable to this article.

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
