# Peer review of "Molecular Dynamics Simulations of Liposomes: Structure, Dynamics, and Applications"

_membranes, 2025, doi:10.3390/membranes15090259_

Round 1
Reviewer 1 Report
Comments and Suggestions for Authors
This review provides a comprehensive summary of molecular dynamics (MD) simulations applied to liposomes, with a focus on their structural and dynamic properties. The authors effectively describe how both coarse-grained (CG) and atomistic MD approaches are used to elucidate key membrane parameters, including area per lipid, bilayer thickness, segmental order, and curvature-induced stress. The roles of cholesterol, PEGylation, and lipid composition in modulating membrane mechanics and drug interactions are thoroughly discussed. The manuscript is well-structured, technically sound, and timely, given the increasing interest in nanocarrier-based drug delivery systems. I enjoyed reading this review. A few minor comments are listed below:
- It would be helpful to include a table summarizing and comparing the key features, advantages, and limitations of CG and AA simulation approaches.
- Consider incorporating more experimental results (e.g., cholesterol-induced changes in bilayer thickness) and directly comparing them with simulation findings.
- The introduction could be improved (current version is very boring) by including more biological context and relevance, especially regarding the functional importance of bilayer properties.
Author Response
Please see the attached response.

Reviewer 2 Report
Comments and Suggestions for Authors
In the review entitled "Molecular Dynamics Simulations of Liposomes: Structure, Dynamics, and Applications," the authors address several important aspects of liposome modeling. The paper begins with a discussion of liposomes and liposomal drug formulations studied using molecular dynamics (MD). It then shifts to the theoretical background, comparing fully atomistic and coarse-grained simulations, including a brief mention of force field (FF) considerations.
This is followed by a section on the role of cholesterol in the structure and function of liposomes, and subsequently, by a discussion on the self-assembly of liposomes using MD. The manuscript then returns to force field discussions (and supposedly includes methods for validation, though this part is not clearly presented). Another thematic shift follows, focusing on curvature and shape effects on liposomal behavior.
Subsequent chapters cover PEGylation, liposome composition and structural diversity (including another partial discussion of cholesterol), drug–lipid interactions and encapsulation mechanisms, and again, cholesterol’s role in liposomal dynamics. The final section presents case studies of marketed liposomal drugs modeled via coarse-grained MD, followed by a brief discussion of current limitations and future directions.
In my opinion, the superficial and repetitive nature of the manuscript makes it unsuitable for publication in its current form. The authors should carefully restructure the review to avoid discussing the same molecules or concepts (e.g., cholesterol) multiple times in different sections. That said, some subchapters are insightful and valuable, such as 13.3. Oversimplified Biological Realism, in which the authors raise an important point regarding the limitations of simulating overly simplified systems that do not fully reflect in vivo (or even in vitro) biological complexity.
Apart from above here are my remarks to selected parts of the MS:
1) Authors should also highlight in Introduction less optimistic view on liposomes. For instance such as latest failure of Thermo-DOX.
2) There exist simulations of full liposome vesicles. Two which I know: 34 nm diameter with MARTINI (dx.doi.org/10.1021/ct500460u) and 20nm diameter with CHARMM (https://dx.doi.org/10.1021/acs.langmuir.0c00475). Authors should check whether there are more and include such aspect in their review.
3) Line 66 - "One of the key insights provided by CG-MD is the effect of curvature on membrane
mechanics." - given examples listed in next sentence, I believe that authors meant dynamics, not mechanics.
4) Line 145 - "Bilayer Thickness: This is measured as the average distance between phosphate headgroups of opposing leaflets (...)" - not necessarly. Sometimes Bilayer thickness (or membrane thickness) it calculated between C2 or C1 atoms. Then it is described by MT_C1-C1 instead of more common MT_P-P. I agree with authors that phosphate-phosphate is most common, but it should be noted that sometimes it is not.
5) Line 153 - SCD is more commonly refered to nowadays as order parameter.
6) Line 164 - One more factor, described in recent theoretical considerations regarding cholesterol in asymmetrical membranes is differential stress. However it is discussed in relate to asymmetrical membranes or membranes with cholesterol after flip-flop, so it might be beyond the scope of the article.
7) Line 207 - The authors state that a moderate concentration of cholesterol is 20–30%, while also claiming that at high concentrations it "can induce phase separation, creating liquid-ordered (Lo) and liquid-disordered (Ld) domains (...)." However, lipid rafts can be formed in systems such as DOPC:Chol:Sphingomyelin in a 1:1:1 ratio, which corresponds to 33% cholesterol. Therefore, the use of the term “high concentration” in this context is misleading and should be clarified or quantified more precisely.
8) Line 286 - The statement "Wang et al. [5] demonstrated temperature-sensitive vesicle formation from various lipid species." is inaccurate. In the cited study, the authors construct a vesicle system into which they load a drug both inside and outside to analyze passive drug loading. Furthermore, the article does not report or explore lipid diffusion; in fact, the term “diffusion” is absent from the entire manuscript. The description of Figure 5 in the current review is misleading. It does not depict vesicle formation but rather shows the equilibration process, illustrating how the drug interacts with membranes of various topologies (e.g., unilamellar vesicles, open vesicle-like structures, etc.). I strongly recommend re-reading the cited article and adjusting the interpretation accordingly.
Author Response
Please see the attached response.

Round 2
Reviewer 2 Report
Comments and Suggestions for Authors
The authors of MS entitled 'Molecular Dynamics Simulations of Liposomes: Structure, Dynamics, and Applications' addressed my remarks and improved the readability of the text. I recommend accepting the MS.